# S100PBP interacts with nucleoporin TPR and facilitates XY crossover formation in mice

Yufan Wu[1,4], Yang Li [ID][1,4], Huan Zhang[1,4], Jingwei Ye[1], Ming Li[1], Jianteng Zhou[1], Xuefeng Xie[1], Hao Yin[1,2,3], Min Chen[1], Gang Yang[1], Suixing Fan [ID][1], Baolu Shi[1], Hanwei Jiang [ID][1✉], Qinghua Shi [ID][1✉] & Hui Ma [ID][1✉]

## Abstract

During meiosis, at least one crossover is selectively generated per pair of homologous chromosomes through homologous recombination to ensure their faithful segregation. The molecular mechanisms controlling meiotic recombination, particularly in XY chromosomes that share a tiny region of homology (i.e., the pseudoautosomal region, PAR), remain poorly understood. Here, we identify S100PBP as a key modulator of both XY and autosomal recombination in mice. *S100pbp*-knockout mice exhibit male infertility and spermatogenesis arrest at meiotic metaphase I, resulting from a drastic reduction in XY crossovers. This failure in XY crossover formation is due to a reduction in TEX11/M1AP-bound recombination intermediates at the PAR. By contrast, disruption of S100PBP significantly increases the number of recombination intermediates and crossovers on autosomes. Co-immunoprecipitation mass spectrometry revealed that S100PBP interacts with the nucleoporin TPR. Furthermore, S100PBP is localized specifically to the nuclear pores of meiocytes, likely in a TPR-dependent manner. These findings demonstrate that S100PBP promotes XY crossover formation while limiting excess autosomal crossovers and shed light on the potential role of nuclear pores in regulating meiotic recombination.

**Keywords** S100PBP; Meiotic Recombination; XY Crossover; Nuclear Pore; TPR
**Subject Categories** Cell Cycle; DNA Replication, Recombination & Repair; Membranes & Trafficking

## Introduction

In meiosis, the generation of crossovers between homologous chromosomes is one of the key events as it is essential for ensuring the proper alignment and segregation of homologous chromosomes at meiotic metaphase I (MMI) (Baudat et al, 2013; Hunter, 2015).

Crossovers are initiated by the generation of programmed double-strand breaks (DSBs) during the leptotene and zygotene stages. The DSB ends are then resected to produce 3' single-stranded DNA (ssDNA) overhangs protected by replication protein A (RPA) proteins, which are later replaced by the recombinase RecA homologs, RAD51 and DMC1 recombinases (Brown and Bishop, 2014; Hinch et al, 2020; Ribeiro et al, 2016). The recombinase-coated ssDNA filaments can recognize and invade the homologous chromosomes to form recombination intermediates which promote the alignment and synapsis of the homologous chromosomes. These recombination intermediates can be stabilized by a cluster of conserved proteins known as the ZMM complex (comprising Zip1-4, Msh4/Msh5, Mer3, and Spo16, initially identified in budding yeast), form double-Holliday junctions, and finally are resolved into crossovers by the endonuclease activity of the MutL homolog heterodimer MLH1-MLH3 (MutLγ) (Cannavo et al, 2020; Dai et al, 2021; Hunter, 2015). If the intermediates fail to be stabilized, they will be processed to generate non-crossover products.

The process of meiotic crossovers is strictly controlled such that crossovers are widely spaced (crossover interference), each homolog pair receives at least one crossover (crossover assurance), and the average number of crossovers per cell is well-controlled (at around 24 per spermatocyte in wild-type mice) (Otto and Payseur, 2019; Wang et al, 2015). Failure in crossover formation would result in infertility due to meiotic arrest or aneuploidy (Handel and Schimenti, 2010). For example, disruption of components in the mouse ZMM subcomplexes, including SHOC1-TEX11-SPO16 (also known as ZZS complex named after the yeast homologs Zip2-Zip4-Spo16) and MSH4-MSH5 (MutSγ), resulted in reduced or no crossover formation, along with incomplete synapsis of varying degrees (Adelman and Petrini, 2008; Santucci-Darmanin et al, 2002; Shinohara et al, 2008; Snowden et al, 2004; Yang et al, 2008). Recently, we found that loss of M1AP, a ZZS-interacting protein, also reduces MSH4 and TEX11 foci, leading to decreased crossovers (Li et al, 2023).

Compared with autosomes, recombination on XY chromosomes is more dedicatedly controlled to assure the formation of a crossover within a tiny homologous segment known as the pseudoautosomal region (PAR). It has been reported that the timing of programmed DSB formation and enzymes catalyzing

[1]Centre for Reproduction and Genetics, First Affiliated Hospital of USTC, Hefei National Laboratory for Physical Sciences at Microscale, School of Basic Medical Sciences, Biomedical Sciences and Health Laboratory of Anhui Province, Institute of Health and Medicine, Hefei Comprehensive National Science Centre, Division of Life Sciences and Medicine, University of Science and Technology of China, Hefei 230027, China. [2]Present address: Department of Molecular Biology, Massachusetts General Hospital, Boston, MA 02114, USA. [3]Present address: Department of Genetics, Harvard Medical School, Boston, MA 02114, USA. [4]These authors contributed equally: Yufan Wu, Yang Li, Huan Zhang. ✉E-mail: jianghanwei.1209@aliyun.com; qshi@ustc.edu.cn; clsmh@ustc.edu.cn

meiotic DSBs at the PAR are different from those of autosomes in mice (Acquaviva et al, 2020; Boekhout et al, 2019; Brick et al, 2012; Kauppi et al, 2011; Kauppi et al, 2012; Lange et al, 2016; Papanikos et al, 2019). Recent findings on mutant mouse models have also shown that disruption of RAD51AP2, a RAD51-interacting protein, or ATF7IP2, a meiosis-specific SETDB1-interacting protein, specifically impairs DSB repair towards crossovers at the PAR, but not on autosomes (Alavattam et al, 2024; Ma et al, 2022; Shao et al, 2023), suggesting that the DSB repair mechanisms leading to crossover formation on XY chromosomes are also differently regulated from those on autosomes.

To unravel new functional genes in meiosis, we conducted in-depth mining of the published single-cell sequencing data of male germ cells and identified a substantial number of genes highly expressed in spermatocytes but without known functions in spermatogenesis. Among these genes, we noted that *S100pbp*, encoding an interacting partner of $Ca^{2+}$-binding S100P protein (Dowen et al, 2005), is highly expressed in the early spermatocytes, implying a potential role in meiosis (Chen et al, 2018). S100PBP has been reported to inhibit the adhesion of pancreatic cancer cells through the S100PBP/CTSZ/RGD αvβ5 pathway (Lines et al, 2012; Lu et al, 2021) and plays a tumor suppressor role (Srivastava et al, 2023). However, whether it is required for spermatogenesis is unknown. In this study, we identify S100PBP as a novel and intriguing regulator of meiotic recombination, which promotes spermatogenesis by facilitating XY crossover formation while constraining autosomal crossover formation. We find that S100PBP localizes to the nuclear pores of meiocytes likely via the interaction with the nucleoporin TPR. Our study elucidates the physiological role of S100PBP and reveals potential regulatory mechanisms involving nuclear pores in mammalian recombination.

## Results

### *S100pbp* is essential for spermatogenesis

To investigate the potential role of *S100pbp*, we first examined its expression using reverse transcription-polymerase chain reaction (RT-PCR) and western blotting. The quantitative polymerase chain reaction (qPCR) assays revealed that *S100pbp* mRNA was detected in various organs examined of adult mice with a predominantly highest expression in testes (Fig. 1A). To confirm the expression of S100PBP protein in testis, we generated an anti-S100PBP antibody that recognizes an epitope consisting of amino acids 19–33 of the mouse protein. S100PBP protein was detected in mouse testes starting from 8 days postpartum (dpp) (Fig. 1B), suggesting a potential role in meiosis.

Next, to explore the physiological function of S100PBP in vivo, we generated *S100pbp*-knockout mice (*S100pbp*$^{-/-}$) using CRISPR/Cas9 technology. *S100pbp*$^{-/-}$ mice carry a 2-base-pair deletion in exon 1 (c.79_80 del) of the *S100pbp* gene, which introduces a premature stop codon (Appendix Fig. S1). Western blotting detected the presence of full-length S100PBP in the wild-type testes, but not in *S100pbp*$^{-/-}$ testes (Fig. 1C). It should be noted that the frameshift allele may produce a truncated protein (p.G27Ffs*4) lacking the C-terminal region or a truncated protein starting at an internal ATG, which, if present, may not be recognized by the anti-S100PBP antibody.

Both *S100pbp*$^{+/-}$ and *S100pbp*$^{-/-}$ mice show no obvious developmental abnormalities and appear healthy. *S100pbp*$^{+/-}$ mice are fertile and capable of breeding normally. Detailed examination of *S100pbp*$^{+/-}$ mice revealed no discernible defects in spermatogenesis. Specifically, there are no significant differences in testis size, testis/body weight ratio, or sperm number when compared to their wild-type littermates (Appendix Fig. S2). However, when we crossed the adult *S100pbp*$^{-/-}$ male mice with wild-type females for a minimum of 3 months, no litters were produced, indicating that the *S100pbp*$^{-/-}$ male mice are infertile (Appendix Table S1). *S100pbp*$^{-/-}$ mice have smaller testes and a significantly reduced testes-to-body weight ratio when compared to those of the control littermates ($7.66 \pm 0.46 \times 10^{-3}$ in controls versus $2.59 \pm 0.10 \times 10^{-3}$ in *S100pbp* knockouts; $n = 4$; $P < 0.0001$; unpaired $t$ test; Fig. 1D,E). While over 10 million spermatozoa per epididymis were detected in the control mice, only a few spermatozoa were seen in the epididymis of *S100pbp*$^{-/-}$ mice (Fig. 1F). Histological analyses of testes from the control mice revealed the presence of numerous spermatogenic cells of all developmental stages. In contrast, the seminiferous tubules of *S100pbp*$^{-/-}$ mice contain many MMI spermatocytes with unaligned chromosomes, and the postmeiotic germ cells are markedly fewer than in the controls. (Fig. 1G). These results suggest that the homozygous frameshift mutation of *S100pbp* in mice causes male infertility and impairs spermatogenesis.

### An increased prevalence of XY univalents in *S100pbp*$^{-/-}$ MMI spermatocytes

At meiotic metaphase I, chiasmata tether each pair of homologous chromosomes, which are referred to as the bivalents, ensuring their alignment at the metaphase plate and the subsequent accurate separation of homologous chromosomes into daughter cells (Baudat et al, 2013; Hunter, 2015). Achiasmatic chromosomes, which are referred to as univalents, would fail to align at the metaphase plate (Carpenter, 1994). Thus, we infer that the unaligned chromosomes observed in *S100pbp*$^{-/-}$ MMI spermatocytes suggested the presence of univalents. To test this possibility, we prepared metaphase chromosome spreads from control and *S100pbp*$^{-/-}$ testes. In control mice, the majority of MMI cells contain 20 bivalents, among which the XY bivalent has a typical appearance of "!" in morphology. However, XY chromosomes were found frequently separated in the mutant MMI cells (Fig. 1H). Analysis of metaphase I chromosomes revealed that, in contrast to 12.86% of MMI spermatocytes containing univalent XY in the control mice, 75.54% of MMI spermatocytes in *S100pbp*$^{-/-}$ mice have univalent XY, while the frequency of univalent autosomes is similar between these two groups (5.42% in controls versus 3.52% in *S100pbp* knockouts) (Fig. 1I). Thus, based on these results, we believe that the unaligned chromosomes observed in *S100pbp*$^{-/-}$ MMI spermatocytes are the univalent XY chromosomes.

### *S100pbp*$^{-/-}$ spermatocytes have increased autosomal crossovers but decreased XY crossovers

The increased frequency of XY separation in *S100pbp*$^{-/-}$ MMI spermatocytes indicates that crossover formation at the PAR is likely abrogated. To test this, we quantified the number of MLH1 foci, which mark sites of crossovers, in spread mid-pachytene

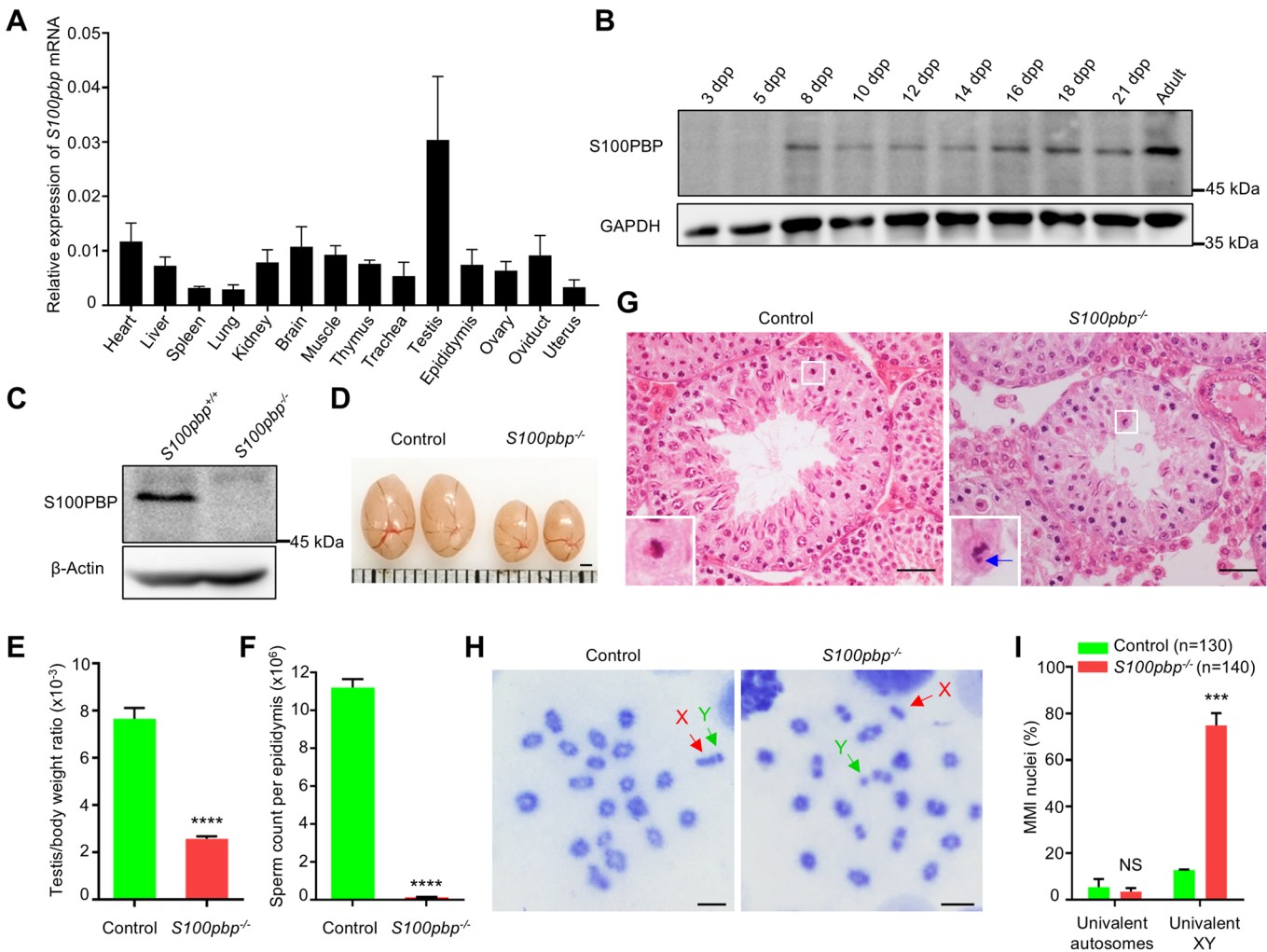

**Figure 1. Meiotic metaphase I arrest with a high prevalence of univalent XY chromosomes in *S100pbp*⁻/⁻ mice.**

(A) Quantitative real-time PCR showing the expression pattern of *S100pbp* in various tissues from adult wild-type mice. Data represent the mean ± SEM of three biological replicates obtained from three different mice for each tissue. (B) Western blot analysis of S100PBP expression in testes from mice at different days postpartum (dpp). GAPDH was used as the loading control. (C) Western blotting with testis lysates from control and *S100pbp*⁻/⁻ mice using the anti-S100PBP antibody. β-Actin was used as the loading control. (D) Representative images of testes from 8-week-old control and *S100pbp*⁻/⁻ mice. Each grid represents 1 mm. (E, F) The ratio of testis/body weight (E) and sperm count per epididymis (F) of 8-week-old control and *S100pbp*⁻/⁻ mice. The data are from at least three biological replicates and represent the mean ± SEM. ****P < 0.0001, two-tailed Student's *t* test. (G) Testicular histology from 8-week-old control and *S100pbp*⁻/⁻ mice. The magnified view of the boxed area is shown in the lower-left corner of the image. The blue arrow indicates the unaligned chromosome in the representative metaphase cell. Scale bars, 50 μm. (H) Representative meiotic metaphase I (MMI) spermatocytes stained with Giemsa. Chromosomes X and Y are indicated. Scale bars, 10 μm. (I) Quantification of the frequencies of nuclei with univalent XY or autosomes in MMI spermatocytes shown in (H). Data represent the mean ± SEM from at least three biological replicates. *n*, the number of MMI cells scored. NS, not significant; ***P = 0.0002; two-tailed Student's *t* test. Source data are available online for this figure.

spermatocytes. The results showed that the frequency of nuclei with an MLH1 focus at the PAR is significantly lower in *S100pbp*⁻/⁻ mice than in control mice (91.88% in controls and 32.65% in *S100pbp* knockouts; Fig. 2A,B), indicating decreased XY crossover formation. Intriguingly, we observed a significant increase of MLH1 foci on autosomes in *S100pbp*⁻/⁻ spermatocytes (25.81 ± 0.28) than in control mice (22.96 ± 0.21) (Fig. 2C) with more autosomal bivalents harboring 2 or 3 MLH1 foci per nucleus (6.79 in *S100pbp* knockouts versus 3.94 in controls, Fig. 2D). To confirm the intriguingly opposite alterations in crossover numbers between the XY and autosomes in *S100pbp* knockouts, we analyzed the staining of MLH3, which interacts with MLH1 to form MutLγ nuclease for

crossover processing, at the mid-pachytene stage. Similar to the observations for MLH1, the frequency of nuclei with an MLH3 focus on the sex chromosomes in *S100pbp*⁻/⁻ spermatocytes is decreased (87.60% in controls and 25.78% in *S100pbp* knockouts; Fig. 2E,F), with more MLH3 foci on autosomes per nucleus (23.33 ± 0.25 in controls versus 25.22 ± 0.29 in *S100pbp* knockouts; Fig. 2G) and autosomal bivalents containing 2 or 3 MLH3 foci per nucleus (Fig. 2H).

In addition, in *S100pbp*⁻/⁻ mice, typical spermatocytes of all four substages of meiotic prophase I (leptotene, zygotene, pachytene, and diplotene) were identified, and their frequencies are all comparable to those in control mice, indicating that the

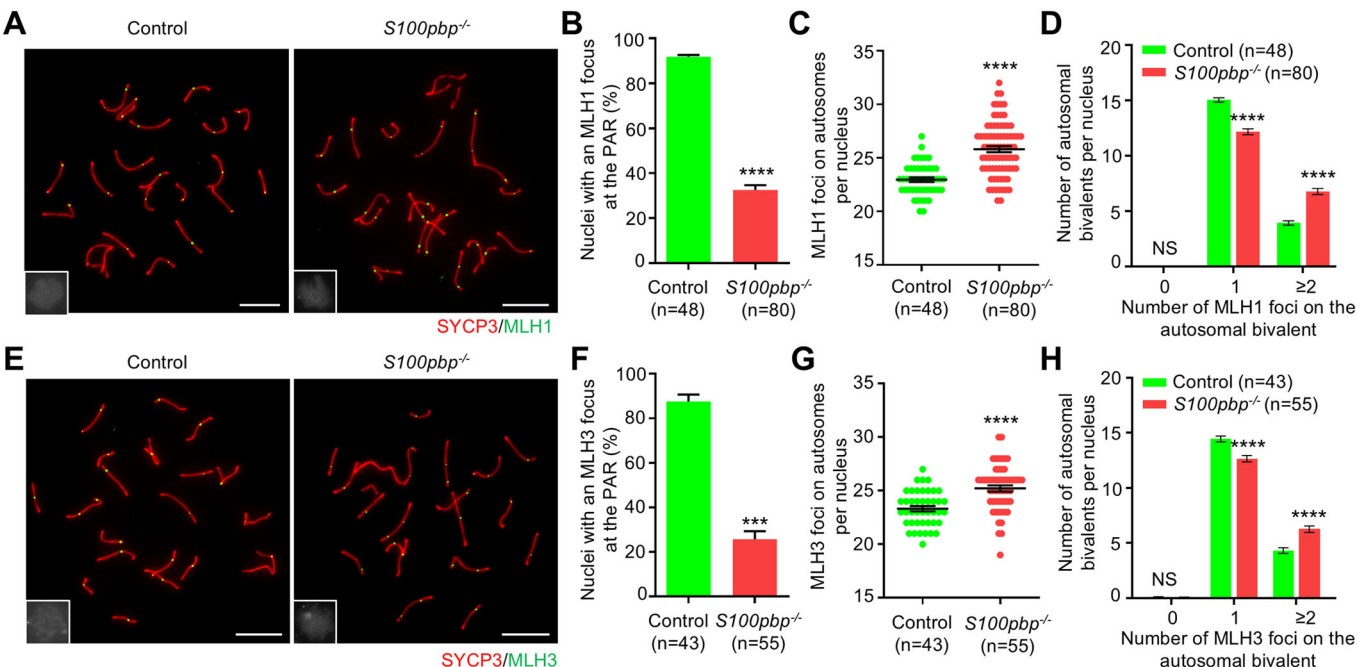

**Figure 2. S100pbp⁻/⁻ spermatocytes show a decrease in XY crossovers while exhibiting an increase in autosomal crossovers.**

(A, E) Representative mid-pachytene spread spermatocytes stained for SYCP3 (red) and MLH1 (green, A) or MLH3 (green, E). Miniaturized H1t staining (gray), shown in the lower-left corner of the overlay images, was used to identify mid-pachytene spermatocytes (H1t-moderate). Scale bars, 10 μm. (B, F) Frequencies of nuclei with an MLH1 focus (B) or an MLH3 focus (F) at the pseudoautosomal region (PAR) of XY chromosomes at the mid-pachytene stage. (C, G) The number of MLH1 foci (C) or MLH3 foci (G) on autosomes per nucleus. (D, H) Number of autosomal bivalents with 0, 1, and ≥2 MLH1 foci (D) or MLH3 foci (H) per nucleus. (B–D, F–H) Data represent the mean ± SEM from at least three biological replicates. n, the number of cells scored. NS, not significant; ***P = 0.0002; ****P < 0.0001; two-tailed Student's t test. Source data are available online for this figure.

S100pbp⁻/⁻ spermatocytes progress normally through meiotic prophase I (Fig. EV1A,B). Similar to control cells, autosomes are fully synapsed and γH2AX staining is restricted to the XY body in the S100pbp⁻/⁻ pachytene cells (Fig. EV1A). Noticeably, there are significantly higher proportions of pachytene cells showing unsynapsed XY and untouching XY PARs in S100pbp mutant mice than those in control mice (Fig. EV1C–E).

These observations together indicate that S100PBP facilitates crossover formation on XY chromosomes, constrains excess crossovers formed on autosomes, and is dispensable for meiotic prophase progression.

### S100pbp-disruption increases the number of genome-wide ZMM-bound recombination intermediates

Previous reports suggest that the elevated number of autosomal MLH1 foci, particularly the higher fractions of bivalents with ≥2 MLH1/MLH3 foci, might be associated with a reduction in crossover interference (Barchi et al, 2008; Girard et al, 2023; Zhang et al, 2014). To explore this, we measured the distances between adjacent MLH1 foci on autosomes with two or more foci, and found that the inter-focus distances in S100pbp⁻/⁻ mice are similar to those of the control mice. This indicates that crossover interference remains unchanged on autosomes of the S100pbp⁻/⁻ spermatocytes (Appendix Fig. S3A).

In addition, there is a well-established correlation between chromosome axis length and the number of crossovers (Wang et al, 2021). We compared the average axis length per nucleus in mid-pachytene

S100pbp⁻/⁻ and control spermatocytes, which revealed no significant difference (Appendix Fig. S3B). These results suggest that the increased autosomal crossovers in S100pbp⁻/⁻ spermatocytes are not attributed to the chromosome axis length either.

Given that changes in crossover numbers could stem from alterations in DSB formation and/or repair processes, we next examined DSB formation by quantifying the numbers of RPA2 and RAD51, which bind the single-stranded DNA overhangs at the DSB sites, in spread spermatocytes. Our analysis revealed that the numbers of RPA2 and RAD51 foci are comparable between S100pbp⁻/⁻ spermatocytes and controls during the leptotene and zygotene stages (Fig. EV2A–D). This indicates that the generation of genomic DSBs remains unaffected by the disruption of S100PBP.

Since DSB formation is not affected, we inferred that the increased crossovers could arise during the formation and processing of recombination intermediates, at the expense of other types of repair. It has been reported that the two ZMM subcomplexes, SHOC1-SPO16-TEX11 (ZZS) and MSH4-MSH5 (MutSγ), bind to and stabilize the recombination intermediates, including the displacement loops (D-loops), single-end invasions (SEIs), and double-Holliday junctions (dHJs), to promote crossover formation (Pyatnitskaya et al, 2019). We thus quantified the dynamics of TEX11, M1AP (an interactor of the ZZS complex), and MSH4 foci on the surface spread of control and S100pbp⁻/⁻ spermatocytes (Fig. 3). In control mice, 143.95 ± 5.02 and 115.48 ± 3.48 TEX11 foci were observed in the late zygotene and early pachytene stages, respectively. We observed a similar trend for TEX11 foci in S100pbp⁻/⁻ spermatocytes, with significantly higher numbers at both the late zygotene (166.35 ± 5.44) and the early pachytene stages

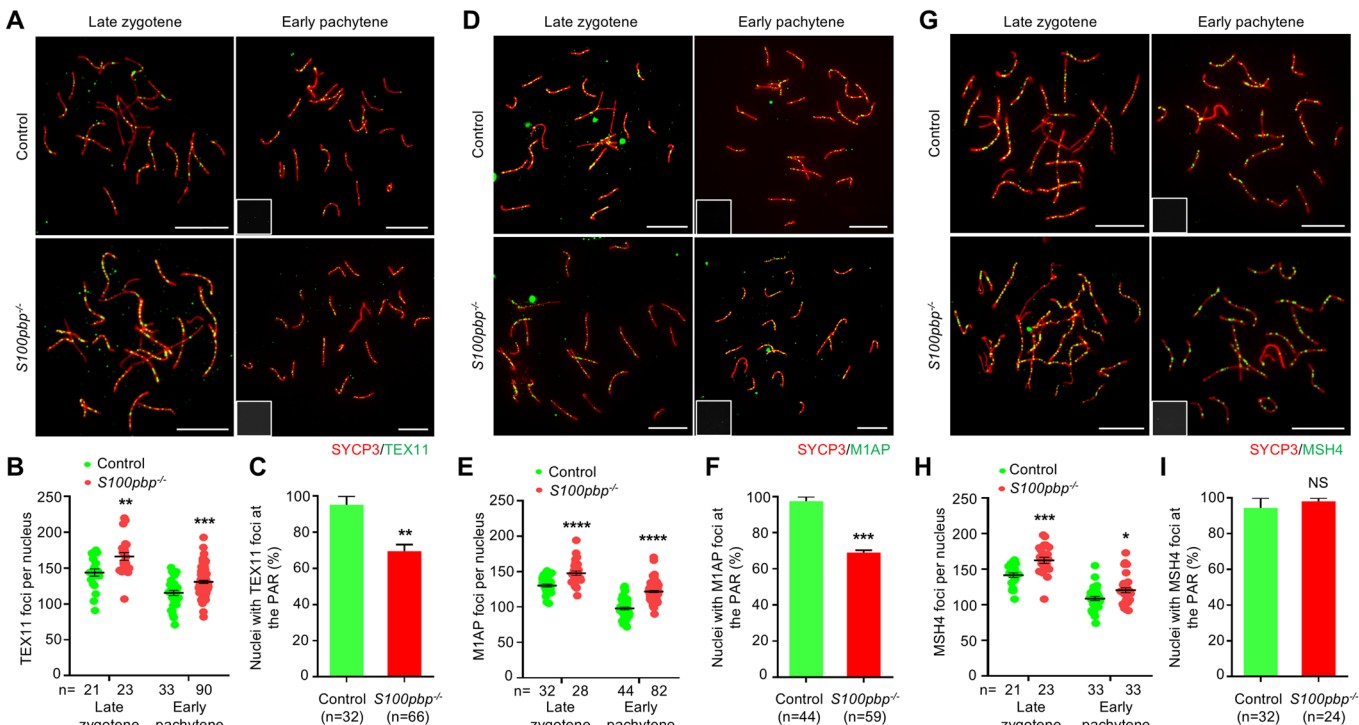

**Figure 3. S100PBP deficiency results in decreased ZZS-bound recombination intermediates at the PAR but increased ZMM-bound recombination intermediates on the autosomes.**

(A, D, G) Immunofluorescence staining with antibodies against SYCP3 (red) and TEX11 (green, A), M1AP (green, D), or MSH4 (green, G), on spermatocyte spreads. Miniaturized H1t staining (gray), shown in the lower-left corner of the overlay images, was used to identify the early pachytene spermatocytes (H1t-negative). Scale bars, 10 μm. (B, E, H) Scatter plots showing the numbers of TEX11 foci (B), M1AP foci (E), or MSH4 foci (H), per nucleus in control and *S100pbp⁻/⁻* spermatocytes at the indicated stages. (C, F, I) Frequencies of nuclei with TEX11 foci (C), M1AP foci (F), or MSH4 foci (I) detected at the PAR in spread early pachytene spermatocytes with touching XY PARs. (B–C, E, F, H, I) Data represent the mean ± SEM from at least three biological replicates. *n*, the number of cells scored. NS, not significant; **P = 0.0044 (B), ***P = 0.0002 (B), **P = 0.0070 (C), ****P < 0.0001 (E), ***P = 0.0006 (F), *P = 0.0104 (H), ***P = 0.0005 (H); two-tailed Student's *t* test. Source data are available online for this figure.

(130.87 ± 2.06) compared to controls (Fig. 3A,B). In addition, we noted an increase in M1AP foci, which facilitates TEX11 recruitment (Li et al, 2023), as well as MSH4 foci in *S100pbp⁻/⁻* spermatocytes (Fig. 3D,E,G,H). This indicates that the *S100pbp* frameshift mutation leads to an increased number of crossover-designated recombination intermediates on the autosomes.

## Decreased number of ZZS-bound recombination intermediates at the PAR of *S100pbp⁻/⁻* spermatocytes

We next examined the generation and repair of DSBs at the PAR, the range of which is defined according to the measured axial length ratios as we previously reported (Ma et al, 2022). In *S100pbp⁻/⁻* mice, the frequencies of nuclei with RPA2 or RAD51 foci at the PAR in early, mid, and late pachytene spermatocytes are all comparable to those in control mice (Fig. EV2E–J). Specifically, no significant differences were observed between nuclei with touching XY PARs and those with untouching XY PARs (Fig. EV2H,J). These results suggest that there are no detectable defects in the generation of PAR DSBs or delays in their repair, in *S100pbp⁻/⁻* mice, regardless of whether the PAR are separated or in contact.

We further analyzed the abundance of recombination proteins at the PAR, focusing on early pachytene spermatocytes with touching XY PARs. We scored the presence of these recombination proteins as a marker of recombination intermediates, following our previous methodology (Ma et al, 2022). In contrast to the genome-wide increase in TEX11 and M1AP foci at the early pachytene stage, their frequencies at the PAR are significantly decreased (TEX11: 95.24 ± 4.76% in controls versus 69.58 ± 3.60% in *S100pbp* knockouts; M1AP: 97.44 ± 2.56% in controls versus 69.02 ± 1.23% in *S100pbp* knockouts; Fig. 3C,F). Intriguingly, the frequency of nuclei with MSH4 foci at the PAR does not differ from that in controls (Fig. 3I). These observations suggested that the recruitment of TEX11 and M1AP is impaired at the PAR, while MSH4 recruitment remains unaffected. Collectively, these findings indicate that *S100pbp* constrains the formation of recombination intermediates committed to crossovers on autosomes but facilitates the XY crossover formation likely by promoting the recruitment of the ZZS complex at the PAR.

## *S100pbp* is dispensable for follicle development and female fertility

*S100pbp* mRNA was detected in fetal ovaries (Fig. 4A), suggesting that it may also play a role in the meiosis of females. To check whether ablation of S100PBP would lead to meiotic defects in oocytes, oocyte spreads from embryos at 17.5 days postcoitus (dpc)

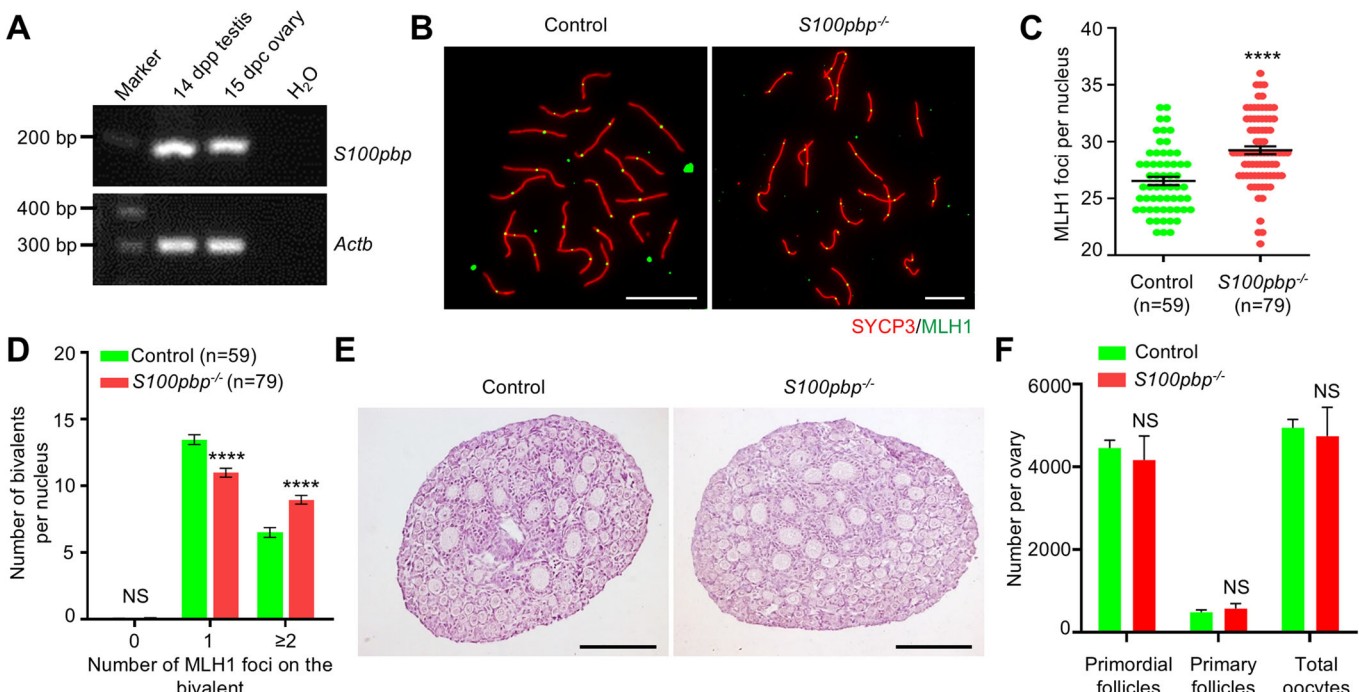

**Figure 4. The number of crossovers is increased while follicle development remains unaffected in *S100pbp*$^{-/-}$ ovaries.**

(A) Reverse transcription-PCR (RT-PCR) analysis of *S100pbp* expression in fetal ovaries (15 days postcoitum (dpc)) and testes (14 dpp) from wild-type mice. *Actb* served as an internal control. (B) Immunofluorescence staining with antibodies against SYCP3 (red) and MLH1 (green) on oocyte spreads from control and *S100pbp*$^{-/-}$ embryos (17.5 dpc). Scale bars, 10 μm. (C) Number of MLH1 foci per nucleus in control and *S100pbp*$^{-/-}$ oocytes. Data represent the mean ± SEM from at least two biological replicates. *n*, the number of cells scored. ****$P < 0.0001$; two-tailed Student's *t* test. (D) The number of bivalents with 0, 1, and ≥2 MLH1 foci per nucleus. Data represent the mean ± SEM from at least two biological replicates. n, the number of cells scored. NS, not significant; ****$P < 0.0001$; two-tailed Student's *t* test. (E) Representative haemotoxylin-stained ovarian sections of 6.5-dpp-old control and *S100pbp*$^{-/-}$ mice. Scale bars, 50 μm. (F) Follicle counts and total oocyte counts per ovary at 6.5 dpp. n, the number of cells scored from at least three biological replicates. NS, not significant; two-tailed Student's *t* test. Source data are available online for this figure.

were prepared for analyzing meiotic recombination. Similar to observations in *S100pbp*$^{-/-}$ spermatocytes, the number of MLH1 foci is increased in *S100pbp*$^{-/-}$ oocytes (Fig. 4B,C), along with more bivalents containing 2, 3, or 4 MLH1 foci per nucleus (Fig. 4D), when compared with those in controls. Nevertheless, it appears that the increased crossover formation does not impede the follicle development in *S100pbp*$^{-/-}$ mice. Analysis of the ovaries from mice at 6.5 dpp of age revealed that the ovary size and morphology of *S100pbp*$^{-/-}$ females are similar to those of control mice (Fig. 4E). Ovarian histological analyses indicated no notable difference in the numbers of total oocytes, primordial follicles, and primary follicles between control and *S100pbp*$^{-/-}$ mice (Fig. 4E,F). Moreover, we did not notice any overt defects in fertility in *S100pbp*$^{-/-}$ female mice. Together, these findings indicate that S100PBP, though not essential for female fertility and follicle development, constrains meiotic recombination in oocytes, which is in congruent with its function in autosomal recombination in spermatocytes.

## S100PBP interacts with TPR and localizes to the nuclear pores in a TPR-dependent manner

To explore the molecular mechanism by which S100PBP promotes meiotic recombination, we performed co-immunoprecipitation (IP) in testicular lysates from 10-week-old wild-type mice using the anti-S100PBP antibody, followed by mass spectrometry (MS) analysis (Fig. 5A). *S100pbp*$^{-/-}$ mice at the same age were used as a parallel negative control. However, it is important to note that differences in cell composition between WT and mutant testes at this age may affect the significance of the observed differences in protein interactions.

Among the 85 candidate S100PBP-interacting proteins (Dataset EV1), TPR, a major structural constituent of the nuclear basket (Krull et al, 2004), ranks first in the number of unique peptides. Western blotting using an anti-TPR antibody validated the presence of TPR in the testicular lysates precipitated by the anti-S100PBP-antibody from the wild-type testes, but not *S100pbp*$^{-/-}$ testes (Fig. 5B). To further confirm the interaction between S100PBP and TPR, we performed co-IP in lysates of HEK-293T cells co-expressing Flag-tagged S100PBP (Flag-S100PBP) and GFP-tagged TPR (GFP-TPR). The result showed that S100PBP interacts with TPR when expressed in cultured cells (Fig. EV3A,B). Interestingly, we noticed that Flag-S100PBP shows a nuclear localization when expressed in HEK-293T cells (Fig. EV3C), consistent with the previous report (Dowen et al, 2005). But when GFP-TPR is co-expressed, Flag-S100PBP anchors to the nuclear envelope and shows a complete co-localization with TPR at the nuclear pores (Fig. EV3D). These observations in cultured cells provided us a hint that S100PBP may localize to the nuclear pores in a TPR-dependent manner.

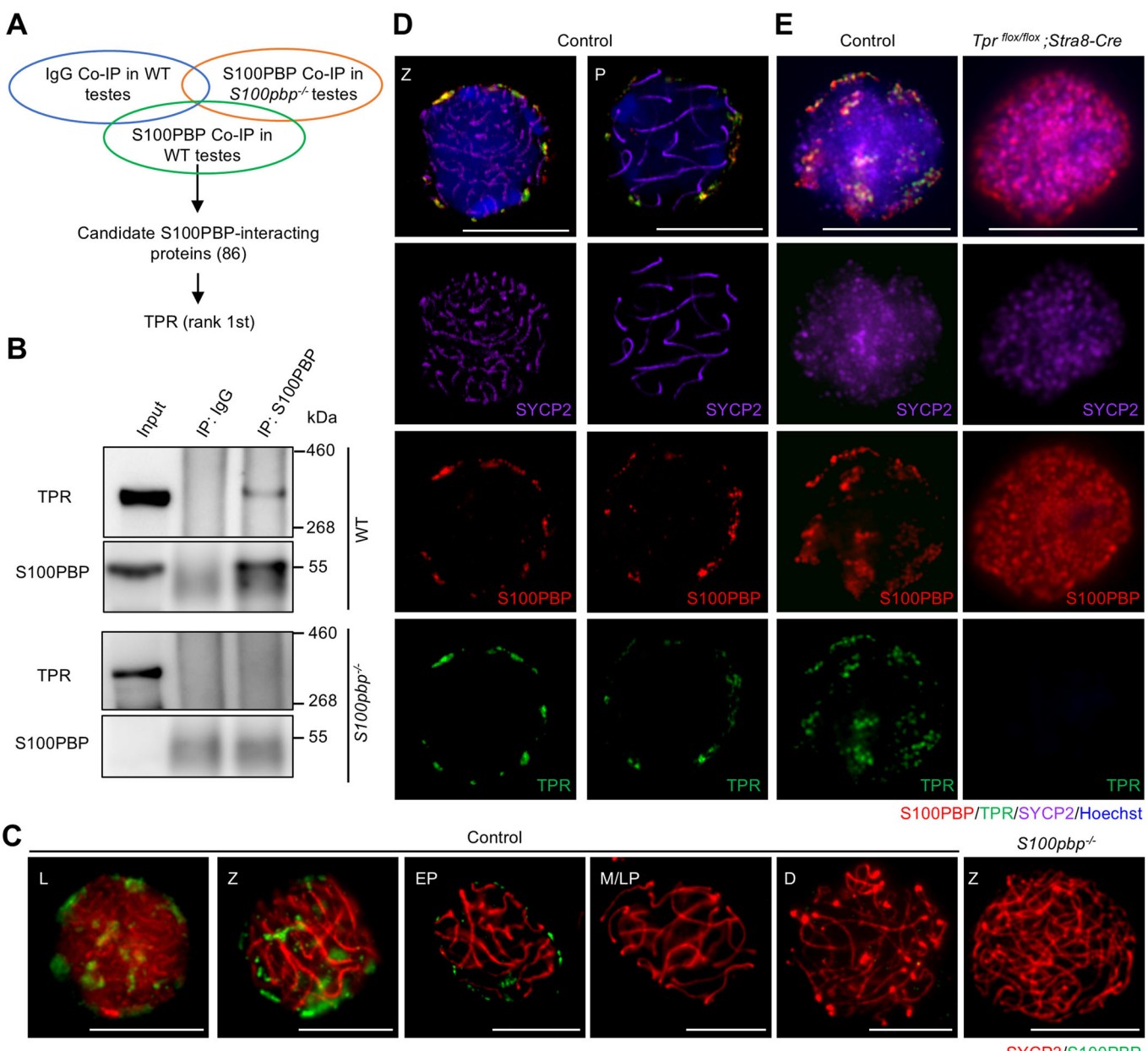

Figure 5. S100PBP interacts with TPR and is localized to the nuclear pores of spermatocytes in a TPR-dependent manner.

(A) Co-immunoprecipitation (co-IP) and mass spectrometry (MS) to identify the S100PBP-interacting proteins. Proteins, which were detected by IP with the anti-S100PBP antibody in the wild-type (WT) testes but absent in groups by anti-rabbit IgG antibody in the wild-type testes and anti-S100PBP antibody in the *S100pbp⁻/⁻* testes, were considered as candidate S100PBP-interacting proteins. The number in the brackets indicates the number of proteins. TPR ranks first in the number of unique peptides. (B) Co-IP using anti-S100PBP antibody with whole-testis lysates of WT mice and *S100pbp⁻/⁻* mice, followed by western blot analyses of S100PBP and TPR. (C) Immunofluorescence staining with antibodies against SYCP3 (red) and S100PBP (green) on spermatocyte smears of control and *S100pbp⁻/⁻* testes. L leptotene, Z zygotene, EP early pachytene, M/LP mid/late pachytene, D diplotene. Scale bars, 10 μm. (D) Confocal imaging of zygotene and pachytene spermatocyte on spermatocyte smears of adult testes from WT mice after immunofluorescence staining of TPR (green), S100PBP (red), and SYCP2 (purple). The nuclei were counterstained with Hoechst 33342 (blue). Z zygotene, P pachytene. Scale bars, 10 μm. (E) Immunofluorescence staining with antibodies against TPR (green), S100PBP (red), and SYCP2 (purple) on spermatocyte smears of control and *Tpr^flox/flox^; Stra8-cre* testes. The nuclei were counterstained with Hoechst 33342 (blue). Scale bars, 10 μm. Source data are available online for this figure.

To further classify the localization of S100PBP, we performed immunofluorescence staining of S100PBP on spermatocyte smears. S100PBP signals were specifically found on the nuclear membrane of leptotene, zygotene, and early pachytene spermatocytes from wild-type mice, and were not observed in other testicular cells or the *S100pbp⁻/⁻* spermatocytes (Fig. 5C). Furthermore, co-staining for S100PBP and TPR revealed a high degree of co-localization in both the spermatocytes and oocytes under the confocal laser scanning microscope (Figs. 5D and EV4).

To confirm whether the localization of S100PBP to the nuclear pores of meiocytes depends on TPR, we tried to knockout *Tpr* specifically in spermatocytes by crossing *Tpr^flox/flox* mice with *Stra8-Cre* transgenic mice in which Cre is activated in germ cells of testes at approximately 3 dpp (Lin et al, 2017) (Fig. EV5A,B). *Tpr^flox/flox; Stra8-cre* mice are infertile and spermatogenesis is arrested at an early spermatogenic stage with few pre-leptotene/leptotene spermatocytes observed (Fig. EV5C–G). Immunofluorescence staining of testicular cell smears confirmed the absence of TPR signals in these spermatocytes, and interestingly, S100PBP loses the nuclear pore localization and becomes dispersed to the nucleus (Fig. 5E).

Taken together, these findings indicate that S100PBP is likely a nuclear pore-associated protein and localizes to the nuclear pores of meiocytes in a TPR-dependent manner.

### The N-terminal amino acid residues 27-94 of S100PBP are crucial for interacting with TPR

To determine which regions of S100PBP interact with TPR, GFP-tagged various deletion mutants of S100PBP, including S100PBP-ΔN (2–155 aa deletion), S100PBP-ΔM (156–269 aa deletion), and S100PBP-ΔC (270–396 aa deletion), were co-expressed with Flag-tagged TPR in HEK-293T cells, and the interaction of these S100PBP mutants with TPR was analyzed by co-IP using GFP-trap magnetic beads (Fig. 6A). S100PBP-ΔM and S100PBP-ΔC retain their interaction with TPR, while S100PBP-ΔN is unable to interact with TPR (Fig. 6B). We next divided the N-terminal amino acid residues (2–155 aa) into three fragments and found that deletion of either 2-50 aa (ΔN1) or 51-100 aa (ΔN2), abolishes the interaction with TPR, whereas the deletion of 101-155 aa (ΔN3) has no adverse effect on the interaction. This suggests that the N-terminal residues 2–100 are essential for the interaction between S100PBP and TPR (Fig. 6A,B).

The N-terminal residues 2–100 do not contain any characterized domain. In silico analysis using the Predictor of Natural Disordered Region (PONDR) (v.VLXT) predicts that S100PBP is enriched in intrinsically disordered regions (IDR), with the highest IDR predictive score value for the region spanning amino acid residues 27-94 (Appendix Fig. S4). To understand whether the interaction between S100PBP and TPR is mediated via this IDR region, we performed co-IP in HEK-293T cells co-expressing GFP-tagged S100PBP-ΔIDR (27-94 aa) and Flag-tagged TPR. Intriguingly, the deletion of IDR abolishes the interaction of S100PBP and TPR (Fig. 6). These results showed that the N-terminal amino acid residues 27-94 of S100PBP are crucial for the interaction between S100PBP and TPR.

## Discussion

Our study uncovers the crucial role of *S100pbp* in mouse meiotic recombination, i.e., facilitating XY crossover formation while limiting excess autosomal crossovers. We also show that S100PBP interacts with TPR and localizes to the nuclear pores of meiocytes likely in a TPR-dependent manner.

The localization of S100PBP suggests that it is more likely to play an indirect role in recombination, than directly processing the recombination intermediates. Indeed, among the candidate interacting proteins of S100PBP identified by MS, we did not find any DSB repair proteins, but the nucleoporin TPR, one of the main components of the nuclear basket (Krull et al, 2004), and some RNA-binding proteins. The nuclear basket, composed of TPR, NUP50, and NUP153, is well-known to regulate the nucleocytoplasmic transport of mRNAs and proteins in somatic cells and also participates in somatic DSB repair (Aksenova et al, 2020; Lee et al, 2020; Simon et al, 2018), but the role of nuclear basket components in mammalian meiosis has not been studied. This could be partially due to that, ablation of these proteins would cause developmental defects before the stage of meiosis, for example, primordial germ cells could not survive after deletion of *Nup50*

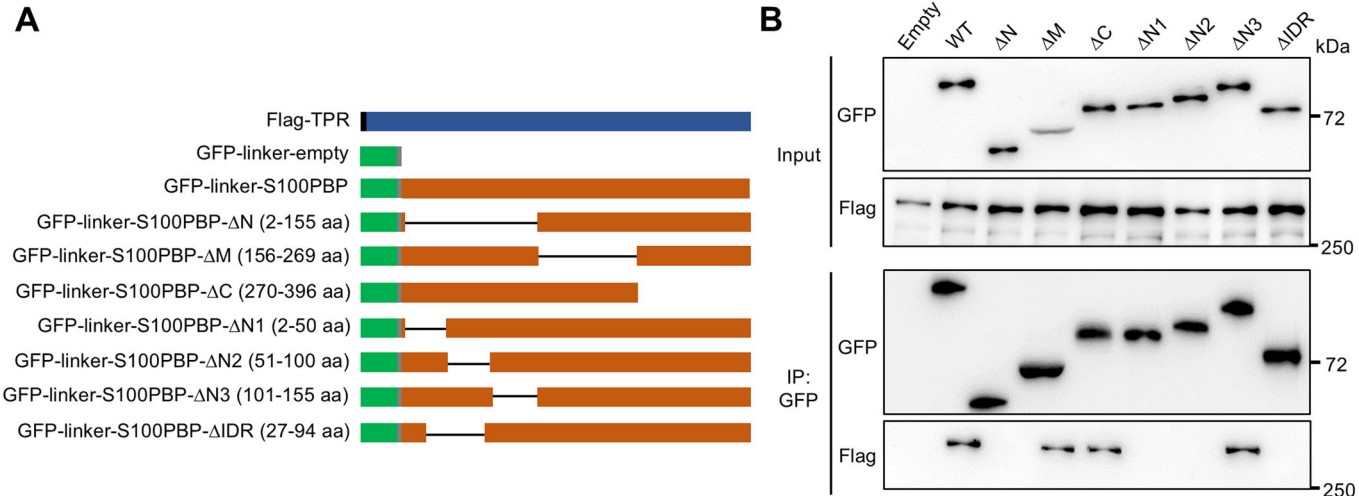

**Figure 6. The N-terminal amino acid residues 27-94 of S100PBP are essential for the interaction with TPR.**

(A) Schematic representation of the mouse proteins: TPR tagged with Flag, and S100PBP and its mutants tagged with GFP. The Flag tag is indicated by the black box, and GFP tags are indicated by the green boxes. (B) Co-IP assays with HEK-293T cells transfected with TPR tagged with Flag and S100PBP or its mutant proteins tagged with GFP. IDR is an intrinsically disordered region. WT wild-type. Source data are available online for this figure.

(Park et al, 2016), as well as what we found (few cells could reach the meiotic stage) after knocking out *Tpr* using the *Stra8-Cre* which is commonly used for conditional gene knockout in meiocytes. Previous reports have shown that even a modest change in the abundance of certain recombinant proteins can influence crossover formation in mice (Reynolds et al, 2013; Yang et al, 2015). It is thus hypothesized that S100PBP may regulate meiotic recombination by mediating the nucleocytoplasmic transport of recombination-related mRNAs and proteins through its interaction with TPR. Therefore, our S100PBP mutant mice could offer valuable insights into the role and underlying mechanisms of nuclear pore complexes in meiosis.

Several mutant mice have been reported to have decreased XY crossovers due to defects in DSB repair, such as mice mutant for *Hfm1*, *Palb2*, *Tex11*, and *M1ap* (Adelman and Petrini, 2008; Guiraldelli et al, 2013; Li et al, 2023; Simhadri et al, 2014; Xie et al, 2022). It should be noted that, in all these mutant mice, the autosomal crossovers are also decreased. In addition, we previously found that in *Rad51ap2* mutant mice, XY crossover is decreased but autosomal crossover formation is not affected (Ma et al, 2022). Recently, it was reported that in mice lacking of ATF7IP2, a SETDB1-interacting protein localizing at the sex body, autosomes have more crossovers but XY crossover is reduced along with meiotic sex chromosome inactivation failure (Alavattam et al, 2024; Shao et al, 2023). They suggested that ATF7IP2 constraints autosomal axis length to influence autosomal crossover and modulates the histone modifications on sex chromosomes to ensure XY crossover, which imply an indirect function in recombination (Shao et al, 2023). In this study, we also found that XY and autosomal recombination present opposing changes after *S100pbp* is disrupted. Interestingly, the increased autosomal recombination in the *S100pbp* mutants is irrelevant to chromosome length, nor any obvious alteration in the formation of sex body, which are different from those in the *Atf7ip2* knockouts. Though the molecular basis of the difference in recombination between sex chromosomes and autosomes in *S100pbp* mutants remains elusive, our study adds new evidence to the speculation that the DSB repair at the XY could be controlled by several different factors and are different from the autosomal recombination.

MSH4 and TEX11 are well-known for their roles in stabilizing intermediates for crossover formation. They show similar foci kinetics, with approximately 60% of TEX11 foci overlapping with MSH4 foci (Yang et al, 2008). Typically, their numbers change similarly when crossover formation is disturbed; for example, both are reduced in *Redic1*$^{-/-}$ mice (Fan et al, 2023) and increased genome-wide in *S100pbp*$^{-/-}$ spermatocytes. Intriguingly, though both MSH4 and TEX11 could localize to PAR (Ma et al, 2022), they are differentially affected in *S100pbp* knockouts. The reduction of the frequency of cells harboring TEX11/M1AP foci at the PAR at the early pachytene stage with touching XY PARs (69.58%/69.02% in *S100pbp*$^{-/-}$ versus 95.24%/97.44% in controls) parallels the increased frequency of untouching XY PARs in *S100pbp*$^{-/-}$ mice from early to mid-pachytene stage (controls: 6.55% and 13.76% in the early and mid-pachytene stage, respectively; *S100pbp*$^{-/-}$: 35.53% and 65.32% in the early and mid-pachytene stage, respectively), implying that the separation of XY chromosomes occurring from early to mid-pachytene stage could be associated with the absence of TEX11/M1AP at the PAR in early pachytene speramtocytes. In contrast, the localization of MSH4 at the PAR appears not adversely affected by the *S100pbp* mutation. Differences in TEX11 and MSH4 localization at the PAR were also reported in *Atf7ip2*$^{-/-}$ pachytene spermatocytes but with MSH4 reduced and TEX11 unaffected (Shao et al, 2023). Based on these observations, we speculate that MSH4 and TEX11 are possibly differently regulated at the PAR, though further experiments are needed to confirm this.

In conclusion, our study revealed that *S100pbp* deficiency results in decreased XY recombination and increased autosomal recombination, offering a new model for exploring the differences in crossover formation between sex chromosomes and autosomes. In addition, our findings suggest that S100PBP's regulatory role in meiotic recombination is likely linked to the nuclear pore complex and add a new dimension to the understanding of the regulatory processes underlying this core event of meiosis.

# Methods

**Reagents and tools table**

| Reagent/resource | Reference or source | Identifier or catalog number |
|---|---|---|
| **Experimental models** | | |
| HEK-293 cells | ATCC | CRL-3216 |
| C57BL/6J | GemPharmatech | Strain No. N000013 |
| ICR | GemPharmatech | Strain No. N000294 |
| *S100pbp*$^{-/-}$ mice | This study | N/A |
| *Tpr-flox* mice | GemPharmatech | Strain No.T008224 |
| *Stra8-Cre* mice | Lin et al, 2017 | N/A |
| **Recombinant DNA** | | |
| Flag-*S100pbp* | This study | N/A |
| Flag-*Tpr* | This study | N/A |
| GFP-*S100pbp* | This study | N/A |
| GFP-*Tpr* | This study | N/A |

| Reagent/resource | Reference or source | Identifier or catalog number |
| --- | --- | --- |
| GFP-*S100pbp*-ΔN | This study | N/A |
| GFP-*S100pbp*-ΔM | This study | N/A |
| GFP-*S100pbp*-ΔC | This study | N/A |
| GFP-*S100pbp*-ΔN1 | This study | N/A |
| GFP-*S100pbp*-ΔN2 | This study | N/A |
| GFP-*S100pbp*-ΔN3 | This study | N/A |
| GFP-*S100pbp*-ΔIDR | This study | N/A |
| **Antibodies** | | |
| Details are available in Appendix Table S2 | | |
| **Oligonucleotides and sequence-based reagents** | | |
| **Primers for qPCR and RT-PCR** | | |
| Mouse *S100pbp*-F | AAGGATTCTGGGGAGGCGAA | |
| Mouse *S100pbp*-R | AGGGCATCTAGCTCTCCCAA | |
| Mouse *Actb* cDNA-F | TAGGCACCAGGGTGTGATGG | |
| Mouse *Actb* cDNA-R | CGTACATGGCTGGGGTGTTG | |
| **Primers for cloning vectors** | | |
| p-N1-F | AGCGGCCGCGACTCTAGATC | |
| P-N1-GFP-linker-R | TCCTGCAGCTCCACCGCTCGACTTGTACAGCTCGTCCATGC | |
| GFP-S100PBP-F | TCGAGCGGTGGAGCTGCAGGAATGACGTGTTCACTCTTGCC | |
| GFP-S100PBP-R | GATCTAGAGTCGCGGCCGCTTCACTGTTGATGGGATGAGA | |
| GFP-S100PBP-ΔN-F | TCGAGCGGTGGAGCTGCAGGAATGTCCTCCAAAGAAACGGAAAA | |
| GFP-S100PBP-ΔN-R | CATTCCTGCAGCTCCACCGCTCGA | |
| GFP-S100PBP-ΔM-F | TTGATAAAGACAAGATCGATTCTGGGGAGGCGAAAGGTGA | |
| GFP-S100PBP-ΔM-R | ATCGATCTTGTCTTTATCAA | |
| GFP-S100PBP-ΔC-F | GACATGCCTTTGACAAGGATTGAAGCGGCCGCGACTCTAG | |
| GFP-S100PBP-ΔC-R | ATCCTTGTCAAAGGCATGTC | |
| GFP-S100PBP-Δ2-50 aa-F | TCGAGCGGTGGAGCTGCAGGAATGTTCAGTTTCACAGAGGAAGA | |
| GFP-S100PBP-Δ2-50 aa-R | CATTCCTGCAGCTCCACCGCTCGA | |
| GFP-S100PBP-Δ51-100 aa-F | GAGAAGAAGATGATGGCCATTTCCTAAAACTACCTCAACT | |
| GFP-S100PBP-Δ51-100 aa-R | ATGGCCATCATCTTCTTCTC | |
| GFP-S100PBP-Δ101-155 aa-F | CAGCAGCTGAAACCCCTGGCTCCTCCAAAGAAACGGAAAA | |
| GFP-S100PBP-Δ101-155 aa-R | GCCAGGGGTTTCAGCTGCTG | |
| GFP-S100PBP-Δ27-94 aa-F | GCAATGCCTCATTTCCTTGGGCAGCTGAAACCCCTGGCTT | |
| GFP-S100PBP-Δ27-94 aa-R | CCAAGGAAATGAGGCATTGC | |
| pEGFP-F | GACTCTGGGGTTCGAAATGA | |
| pEGFP-R | TCATTTCGAACCCCAGAGTC | |
| P-MCS-Flag-R | ATCGTCATCGTCTTTGTAATCCATGGTGGCGACCGGTGGATCCC | |

| Reagent/resource | Reference or source | Identifier or catalog number |
|---|---|---|
| Flag-S100PBP-F | GATTACAAAGACGATGACGATAAAATGACGTGTTCACTCTTGCC | |
| Flag-S100PBP-R | GATCTAGAGTCGCGGCCGCTTCACTGTTGATGGGATGAGA | |
| GFP-TPR-F | TCGAGCGGTGGAGCTGCAGGAATGACCTCTGGTGGCTCG | |
| GFP-TPR-R | GATCTAGAGTCGCGGCCGCTTTAATTAATATTCCCTCTATTTATTCCTCCTCTTCCT | |
| TPR-HF1 | AGACCTGCGCTCACAAAACA | |
| TPR-HR1 | TGTTTTGTGAGCGCAGGTCT | |
| TPR-HF2 | GAAGCAAGAAGTCTCCAGGA | |
| TPR-HR2 | TCCTGGAGACTTCTTGCTTC | |
| Flag-TPR-F | GATTACAAAGACGATGACGATAAAATGACCTCTGGTGGCTCG | |
| Flag-TPR-R | GATCTAGAGTCGCGGCCGCTTTAATTAATATTCCCTCTATTTATTCCTCCTCTTCCT | |
| **Primers for sgRNA transcription** | | |
| *S100pbp*-knockout (KO) -sgRNA-F | GAAATTAATACGACTCACTATAG GGAGATCTTCATCCAAGGAACCCCAGTTTTAGAGC | |
| sgRNA-R | AAAAAAGCACCGACTCGGTG | |
| **Primers for mouse genotyping** | | |
| *S100pbp*-KO-F | AGCCTCACCAACCCAAATCA | |
| *S100pbp*-KO-R | CTGGTCCCAAGCTGTACAAA | |
| *Stra8*-cre-F | AACATTTGGGCCAGCTAAAC | |
| *Stra8*-cre-R | CATCCTTAGCGCCGTAAATC | |
| *Tpr*-cKO-5'-F | ACCTGTAGTTAGTCATAGACAGCTGG | |
| *Tpr*-cKO-5'-R | TGCTTTCACACTTAGAAACCGC | |
| *Tpr*-cKO-3'-F | TACTTCCGGGATAGGTAAGCATTG | |
| *Tpr*-cKO-3'-R | CTGTTTAAGACCCTTTTATACCACTG | |
| **Chemicals, enzymes, and other reagents** | | |
| Anti-GFP nanobody magarose beads | Alpalifebio | KTSM1334 |
| Bouin's solution | Sigma | HT10132 |
| Bromophenol blue | Sinoreagent | 71008060 |
| ClonExpress MultiS One Step Cloning Kit | Vazyme | C113-02 |
| DMEM | VivaCell | C3113-0500 |
| Dithiothreitol (DTT) | Solarbio | D0632 |
| Eosin Y | Solarbio | E8080 |
| FastStart Universal SYBR Green Master (ROX) | Roche | 04913850001 |
| Fetal Bovine Serum (FBS) | GIBCO | 16000-044 |
| Giemsa stain | Solarbio | DM0002 |
| Glycerol | BIO BASIC INC | GB0232 |
| HCL | Sinoreagent | 10011018 |
| Hematoxylin | Sangon | A426825 |
| Lipofectamine 3000 | Invitrogen | L3000015 |
| NaCl | Sinoreagent | 10019318 |
| Opti-MEM | GIBCO | 31985-070 |
| Paraformaldehyde (PFA) | Sinoreagent | 80096618 |
| Paraplast | Leica | 39601095 |

| Reagent/resource | Reference or source | Identifier or catalog number |
|---|---|---|
| Peniciuin-streptomycin | GIBCO | 15140-122 |
| Phanta Max Master Mix | Vazyme | P525 |
| MEGA shortscript T7 kit | Thermo Fisher | AM1354 |
| Phenylmethanesulfonyl fluoride (PMSF) | Thermo Fisher Scientific | 36978 |
| PrimeScript RT reagent kit | Takara | RR047A |
| Protein A/G agarose beads | Santa Cruz | sc-2003 |
| Sodium dodecyl sulfate (SDS) | Sinoreagent | 30166428 |
| Sucrose | Sigma | S5390 |
| Tris-base | Biofroxx | 1115GR500 |
| Trisodium citrate dihydrate | Sinoreagent | 10019418 |
| Triton X-100 | Sangon | A600198-0500 |
| TRIzol | Takara | 9109 |
| VECTASHIELD | Vector | H-1000 |
| Software | | |
| Prism 8 | Graphpad | RRID:SCR_002798 |
| Image-Pro Plus | MediaCybernetics | RRID:SCR_016879 |

## Mice

*S100pbp* mutant mice were generated using CRISPR/Cas9 technology as we described previously (Jiang et al, 2017; Li et al, 2023). Briefly, the sgRNAs were designed targeting the exon 1 (for generating $S100pbp^{-/-}$ mice) and transcribed in vitro using MEGA shortscript T7 kit (Thermo Fisher Scientific, AM1354). The obtained sgRNAs were microinjected into the zygotes of C57BL/6J mice together with Cas9 mRNA, and the zygotes were then transferred into the oviducts of pseudopregnant ICR female mice. Sanger sequencing with genomic DNA extracted from mouse toes was performed to detect the mutations in the mice of $F_0$ generation. The founder mice carrying the mutation of interest in *S100pbp* were backcrossed with wild-type C57BL/6 mice for at least one generation and the obtained heterozygous mutant mice were intercrossed to produce homozygous mice. *Tpr-flox* mice (Strain NO.T008224) were purchased from GemPharmatech. *Stra8-Cre* mouse was gifted by Prof. Ming-Han Tong (Lin et al, 2017). All mouse experiments were conducted using adult mice aged between 8 and 12 weeks (unless specifically stated) and were carried out following the guidelines approved by the Institutional Animal Care Committee of USTC (approval number: USTCA-CUC25120122053). The primers for genotyping and sgRNAs used in this study are listed in Reagents and Tools Table.

## RNA isolation, RT–PCR, and qPCR

Total RNAs were extracted using TRIzol reagents (Takara, 9109) and cDNAs were synthesized from total RNAs using the PrimeScript RT reagent kit (Takara, RR047A) according to the manufacturer's protocol. Phanta Max Master Mix (Vazyme, P525) was used for subsequent PCR under the following conditions: 3 min at 95 °C, followed by 35 cycles of 15 s at 95 °C, 15 s at 60 °C, and 30 s at 72 °C. The qPCR was performed with FastStart Universal SYBR Green Master (ROX) (Roche,

04913850001) using a StepOne Real-Time PCR System (Applied Biosystems). The qPCR reactions were performed under the following conditions: 10 s at 95 °C, 40 cycles of 10 s at 95 °C, and 30 s at 60 °C. *Actb* was used as the internal control. The sequences of the primers are listed in the Reagents and Tools Table.

## Histology

Mouse testicular tissues were harvested and fixed in Bouin's solution. Mouse ovaries were harvested and fixed in modified Davidson's fluid overnight. After embedding in paraffin, the tissue was sectioned at 5 μm. Paraffin-embedded ovaries were serially sectioned and mounted on slides for counting the number of follicles. Hematoxylin and eosin (H&E) staining of the testes and hematoxylin staining of the ovaries were performed as we described previously (Ma et al, 2022).

## Diakinesis/metaphase I chromosome spreading

Chromosome spreads of meiotic diakinesis/metaphase cells were prepared, as previously described, and stained with Giemsa (Jiang et al, 2017).

## Spermatocyte chromosome spreads and immunofluorescence staining

Spermatocyte chromosome spreading and subsequent immunofluorescence staining was performed as previously reported (Jiang et al, 2014; Ma et al, 2022). Information about antibodies is available in Appendix Table S2.

## Oocyte chromosome spreads

Fetal ovaries were collected from embryos at 17.5 dpc and oocyte spread as previously described (Hwang et al, 2018; Li et al, 2023).

Briefly, ovaries were placed in the hypoextraction buffer, containing 30 mM tris (pH 8.2), 50 mM sucrose, 17 mM trisodium citrate dihydrate, 5 mM EDTA, 2.5 mM dithiothreitol (DTT), and 1 mM PMSF, for 15 min, then transferred into 100 mM sucrose, and teased apart with two needles. Twenty microliters of the cell suspension were added onto slides mounted with 200 μL of 1% paraformaldehyde containing 0.15% Triton X-100, evenly spread on the slides, and then incubated in a humid chamber for 2 h at room temperature. The slides were then air-dried and immunofluorescence staining was subsequently performed.

## Co-IP and mass spectrometry

The mouse testes were lysed in IP buffer (50 mM Tris-HCl pH 7.5, 150 mM NaCl, 0.5% Triton X-100, and 2.5 mM EDTA) supplemented with 1 mM PMSF. The lysates were sonicated for 15 cycles (2 s on/off) with 12% pulses and centrifuged at $15,000 \times g$ at 4 °C for 15 min. The supernatant was incubated with precleared Protein A/G agarose beads (Santa Cruz, sc-2003) and 2 μg anti-S100PBP antibody (epitope: residues 19–33 of mouse protein; customizedly produced by ABclonal) or rabbit IgG antibody (ABclonal, AC005). After incubation at 4 °C for 8 h, the agarose beads were washed in IP buffer three times and stored at −80 °C. Western blotting using the anti-S100PBP antibody, as well as silver staining, was performed to assess the quality of Co-IP. The IPed samples were subjected to mass spectrometry (MS) at the National Center for Protein Science Shanghai. The proteins identified by MS and the candidate proteins interacting with S100PBP are available in Dataset EV1.

## Plasmids

To construct vectors expressing Flag-tagged or GFP-tagged mouse S100PBP protein and mouse TPR protein, the coding sequences of *S100pbp* and *Tpr* were amplified from mouse testis cDNA by PCR and ligated with the backbone from pEGFP-N1 vectors using the ClonExpress MultiS One Step Cloning Kit (Vazyme, C113) according to the manufacturer's instructions. The mutated variants, including GFP-*S100pbp*-ΔN, GFP-*S100pbp*-ΔM, GFP-*S100pbp*-ΔC, GFP-*S100pbp*-ΔN1, GFP-*S100pbp*-ΔN2, GFP-*S100pbp*-ΔN3, and GFP-*S100pbp*-ΔIDR, were constructed based on GFP-*S100pbp*. All vectors were verified by Sanger sequencing. The sequences of the primers for plasmid construction are listed in the Reagents and Tools Table.

## Cell culture, transfection, and immunofluorescence

HEK-293T cells (ATCC, CRL-3216) were cultured in high-glucose Dulbecco's modified Eagle's medium (DMEM) supplemented with 10% FBS (Gibco, 16000-044), 100 U/mL penicillin, and 100 mg/mL streptomycin (Gibco, 15140-122) and maintained at 5% $CO_2$. The cells were tested negative for mycoplasma contamination. Cells were passaged 2–3 times after thawing and transfected at 70%-90% confluency. Transfection of plasmids was performed using Lipofectamine 3000 (Invitrogen, L3000015) according to the manufacturer's instructions.

Twenty-four hours after cell transfection, cells were fixed in 4% paraformaldehyde followed by immunofluorescence. The primary antibodies were incubated at 4 °C overnight and followed with secondary antibodies for 1 h at 37 °C. Finally, the slides were

mounted with Vectashield medium (Vector Laboratories, H-1000). The antibodies used for cell immunofluorescence are listed in Appendix Table S2.

## Co-IP in cultured cells

Forty-eight hours after cell transfection, cells were harvested for protein extraction and lysed in IP buffer supplemented with 1 mM PMSF. The obtained protein lysates were incubated with anti-GFP nanobody magarose beads (Alpalifebio, KTSM1334) under gentle rotation at 4 °C for 6 h. After washing with IP buffer three times, the beads were boiled in 1× SDS sample buffer (100 mM Tris-HCl pH 7.4, 2% SDS, 15% glycerol, 0.1% bromophenol blue and 5 mM DTT) for 10 min, followed by western blotting analyses.

## Cell smear

Testes from adult mice and ovaries from embryos (16.5 dpc) were harvested. The tissues were placed in PBS supplemented with 10% FBS and teased apart to release cells. Cell suspension was added to one end of the glass slide and spread using a glass capillary tube gently. After air-drying, the slide was fixed in 4% PFA for 10 min and permeabilized in 0.2% PBST for 30 min, followed by immunofluorescence staining. Confocal images were captured using a Nikon C2 Plus Confocal Laser Scanning Microscope.

## Statistical analysis

The results of each group were presented as mean ± SEM. All statistical analyses comparing *S100pbp* mutant mice with control mice (wild-type or heterozygous littermates) were performed using a two-tailed unpaired Student's *t* test, as specified in the figure legends. The difference was considered significant when the $P$ value was <0.05. No blinding, randomization, and sample size estimations were performed.

# Data availability

This study includes no data deposited in external repositories.

The source data of this paper are collected in the following database record: biostudies:S-SCDT-10_1038-S44319-025-00391-y.

# Peer review information

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

## Acknowledgements

The authors thank all the participants for their cooperation. The authors thank Prof. Ming-Han Tong for generously gifting us the *Stra8-Cre* mice. The authors thank Prof. Mengcheng Luo for generously gifting us the anti-MLH3 antibodies. The authors thank the Mass Spectrometry System at the National Facility for Protein Science in Shanghai (NFPS) for mass spectrometry testing. The authors also thank the Bioinformatics Center of the University of Science and Technology of China, School of Life Sciences, for providing supercomputing resources. This work was supported by the National Natural Science Foundation of China (32270901 to HM), the National Key Research and Developmental Program of China (2022YFC2702600 to QS and HM), the National Natural Science Foundation of China (32470915 to BS and U21A20204 to QS), the National Key Research and Developmental Program of China (2022YFA0806303 to HZ), and the Global Select Project (DJK-LX-2022010 to QS) of the Institute of Health and Medicine, Hefei Comprehensive National Science Center, and the USTC Research Funds of the Double First-Class Initiative (YD9100002034 to QS).

## Author contributions

**Yufan Wu**: Formal analysis; Validation; Investigation; Visualization; Writing—original draft. **Yang Li**: Formal analysis; Validation; Investigation; Visualization; Writing—review and editing. **Huan Zhang**: Funding acquisition; Methodology. **Jingwei Ye**: Investigation; Methodology. **Ming Li**: Investigation; Methodology. **Jianteng Zhou**: Formal analysis; Methodology. **Xuefeng Xie**: Investigation; Methodology. **Hao Yin**: Investigation; Methodology. **Min Chen**: Investigation; Methodology. **Gang Yang**: Investigation; Methodology. **Suixing Fan**: Investigation; Methodology. **Baolu Shi**: Investigation; Methodology. **Hanwei Jiang**: Supervision; Investigation. **Qinghua Shi**: Conceptualization; Supervision; Funding acquisition; Writing—review and editing. **Hui Ma**: Conceptualization; Supervision; Methodology; Funding acquisition; Writing—original draft; Writing—review and editing.

Source data underlying figure panels in this paper may have individual authorship assigned. Where available, figure panel/source data authorship is listed in the following database record: biostudies:S-SCDT-10_1038-S44319-025-00391-y.

## Disclosure and competing interests statement

The authors declare no competing interests.

# Expanded View Figures

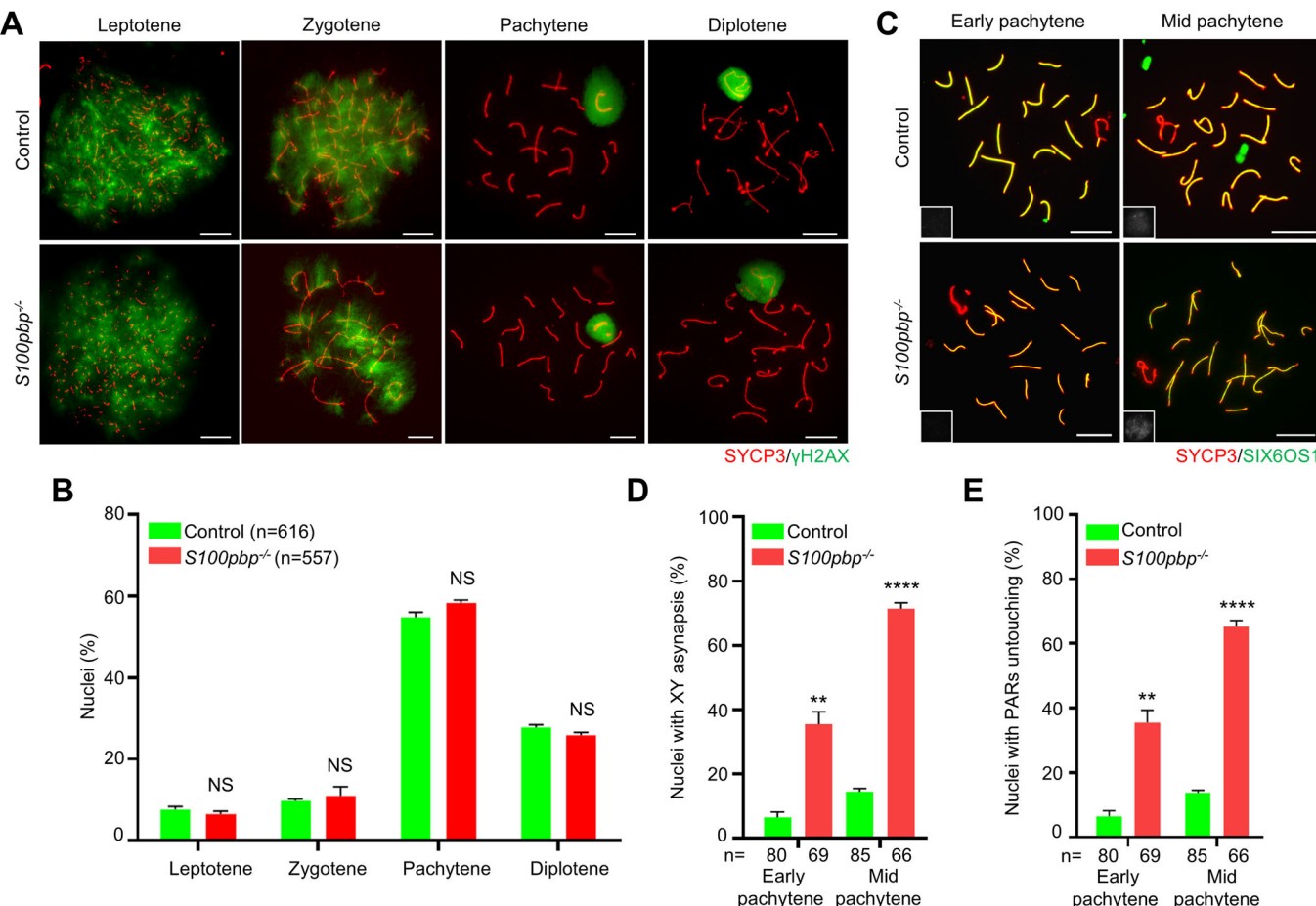

**Figure EV1.  The progression of meiotic prophase I analysis of *S100pbp*⁻/⁻ mice.**

(**A**) Immunofluorescence staining of SYCP3 (red) and γH2AX (green) on spermatocyte spreads from 8-week-old control and *S100pbp*⁻/⁻ mice. Scale bars, 10 μm. (**B**) The percentages of spermatocytes of each substage of meiotic prophase I in 8-week-old control and *S100pbp*⁻/⁻ mice. Data represent the mean ± SEM from at least three biological replicates. *n*, the number of cells scored. NS, not significant; two-tailed Student's *t* test. (**C**) Representative spread spermatocytes stained for the lateral element (SYCP3, red) and the central element (SIX6OS1, green) at the early, mid or late pachytene stages. Miniaturized H1t staining (gray), shown in the lower-left corner of the overlay images. Scale bars, 10 μm. (**D, E**) Frequencies of nuclei with XY asynapsis (**D**) and nuclei with XY PARs untouching (**E**). Data represent the mean ± SEM from at least three biological replicates. *n*, the number of cells scored. **$P = 0.0022$; ****$P < 0.0001$; two-tailed Student's *t* test.

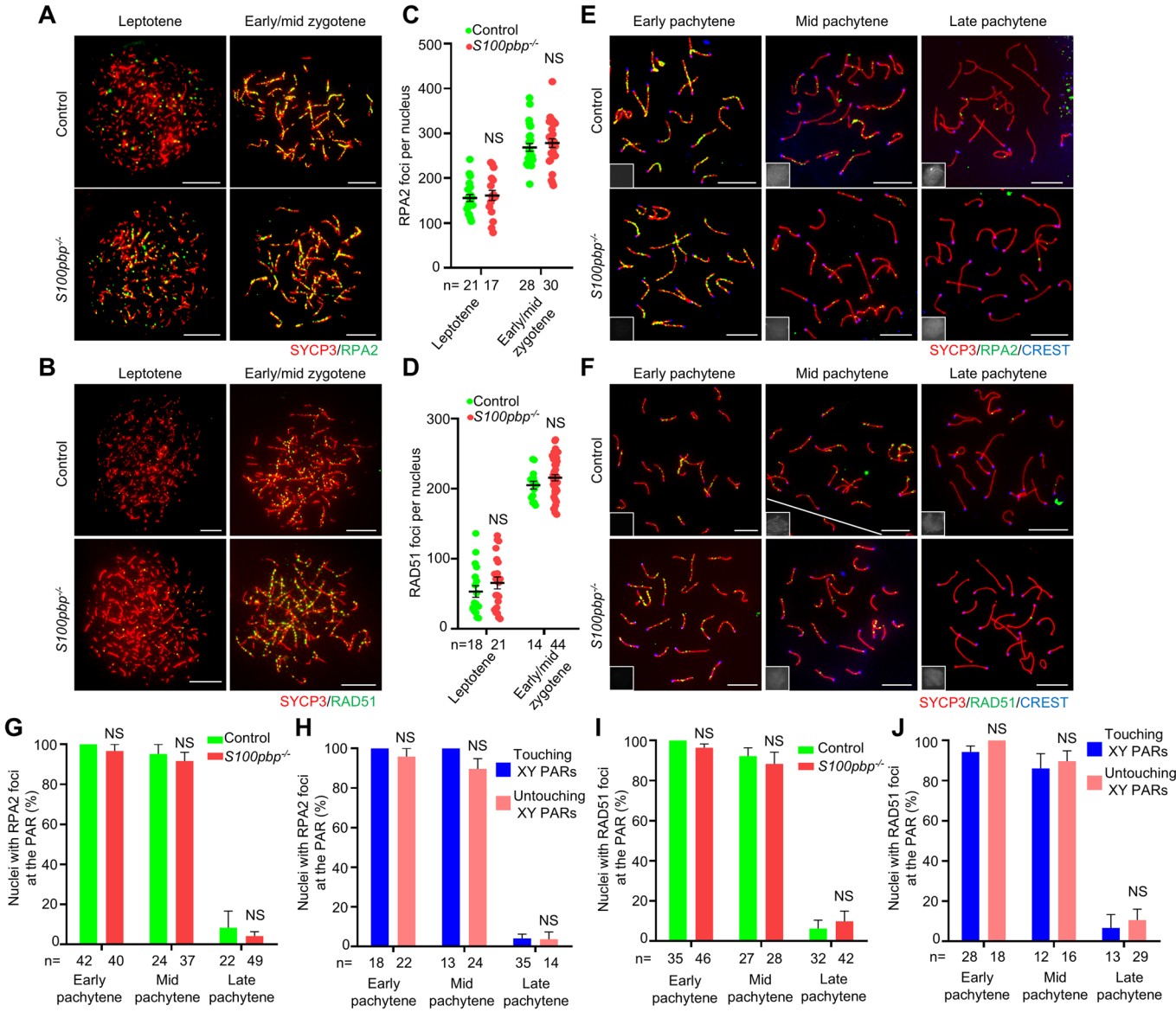

**Figure EV2. The dynamics of RPA2 and RAD51 foci in *S100pbp*<sup>−/−</sup> spermatocytes.**

(A, B) Immunofluorescence staining with antibodies against SYCP3 (red) and RPA2 (green, A) or RAD51 (green, B) on spermatocyte spreads. Scale bars, 10 μm.
(C, D) Scatter plots showing the number of RPA2 foci (C) and RAD51 foci (D) per nucleus in control and *S100pbp*<sup>−/−</sup> spermatocytes at the indicated stages. *n*, the number of cells scored from at least three biological replicates. NS, not significant; two-tailed Student's *t* test. (E, F) Immunofluorescence staining with antibodies against SYCP3 (red), CREST (blue), and RPA2 (green, E) or RAD51 (green, F) on spermatocyte spreads. Miniaturized H1t staining (gray) images are shown in the lower-left corner of the overlay images. Scale bars, 10 μm. (G, I) Frequencies of nuclei with RPA2 foci (G) and RAD51 foci (I) were detected at the pseudoautosomal region (PAR) in spread early, mid, and late pachytene spermatocytes. (H, J) Frequencies of nuclei with RPA2 foci (H) and RAD51 foci (J) detected at the PAR were compared between nuclei with touching XY PARs and those with untouching XY PARs in *S100pbp*<sup>−/−</sup> mice. For (C), (D), and (G–J), data represent the mean ± SEM from at least three biological replicates. *n*, the number of cells scored. NS, not significant; two-tailed Student's *t* test.

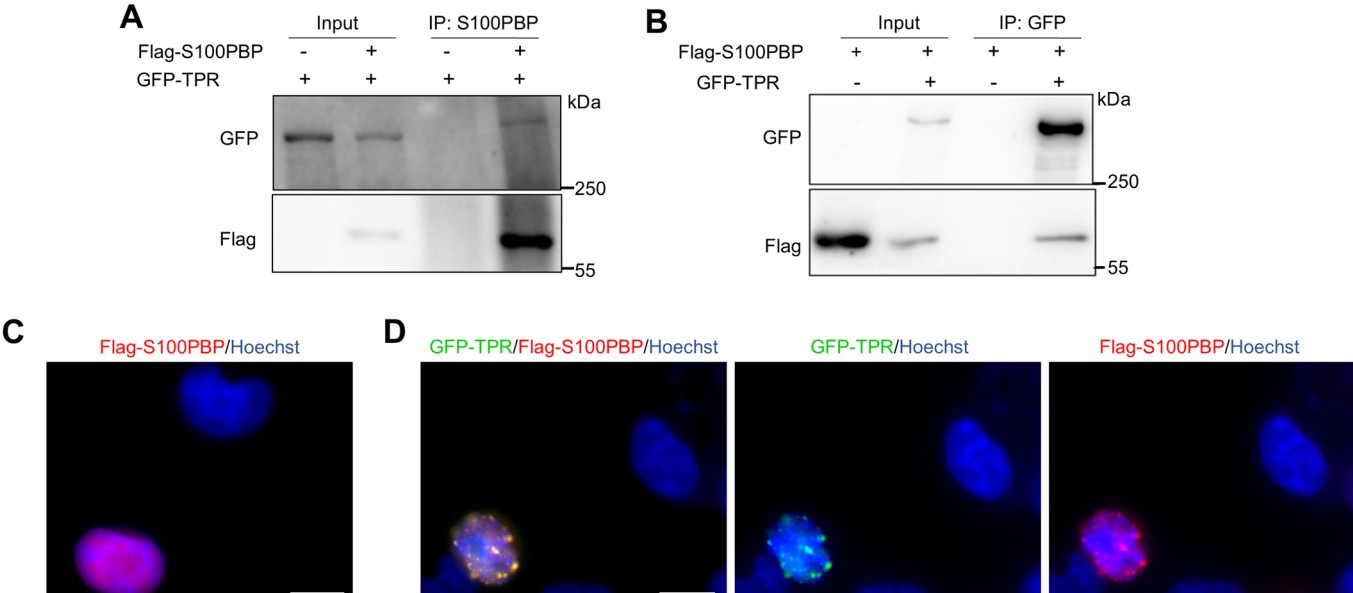

**Figure EV3.  Co-immunoprecipitation (Co-IP) and co-immunofluorescence staining confirmed the interaction and co-localization between S100PBP and TPR exogenously expressed in cultured HEK-293T cells.**

(A, B) Co-IP was performed using an anti-S100PBP antibody (**A**) and an anti-GFP antibody (**B**) in HEK-293T cells that were exogenously expressing S100PBP (with an N-terminal Flag tag) and TPR (with an N-terminal GFP tag), followed by western blotting with the anti-Flag and anti-GFP antibodies. (C, D) Immunofluorescence staining with antibodies against Flag (red) and GFP (green) antibodies in HEK-293T cells exogenously expressing Flag-S100PBP (**C**), or Flag-S100PBP and GFP-TPR (**D**). The nuclei were counterstained with Hoechst 33342 (blue). Scale bars, 10 μm.

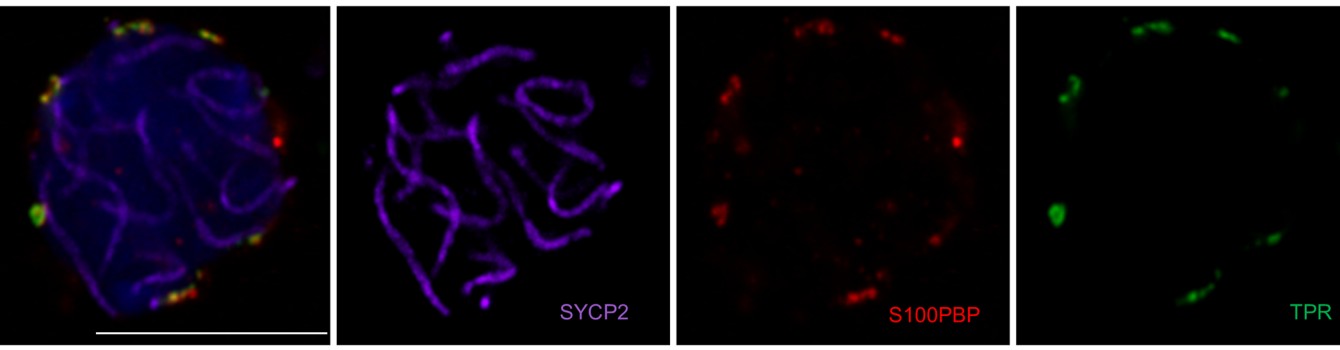

**Figure EV4. S100PBP and TPR are co-localized in oocytes.**

Representative confocal imaging of zygotene oocytes on the oocyte smear of fetal ovaries from wild-type mice (16.5 dpc) after immunofluorescence staining for TPR (green), S100PBP (red) and SYCP2 (purple). The nuclei were counterstained with Hoechst 33342 (blue). Scale bars, 10 μm.

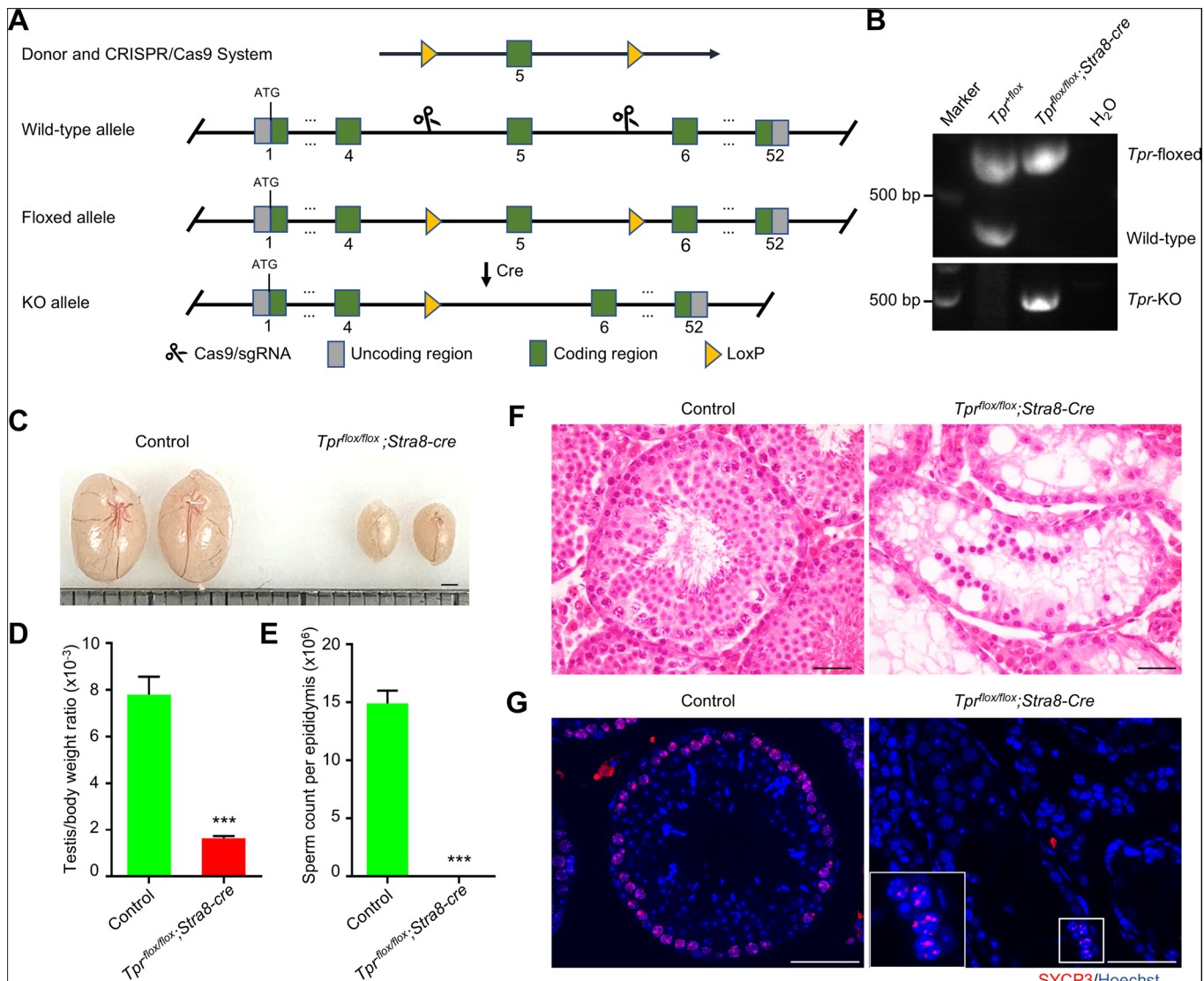

**Figure EV5. Generation and spermatogenic analysis of *Tpr^flox/flox*;*Stra8-cre* mice.**

(A) The strategy to generate *Tpr^flox/flox*;*Stra8-cre* mice. (B) PCR analysis of genomic DNA confirmed the *Tpr* mutation in *Tpr^flox/flox*;*Stra8-cre* testes. KO, knockout.
(C) Representative images of testes from 8-week-old control and *Tpr^flox/flox*;*Stra8-cre* mice. Each grid represents 1 mm. (D, E) The ratio of testis/body weight (D) and sperm count per epididymis (E) of 8-week-old control and *Tpr^flox/flox*;*Stra8-cre* mice. The data are from at least three biological replicates and represent the mean ± SEM.
***$P = 0.0002$ (D, E); two-tailed Student's $t$ test. (F) Testicular histology from 8-week-old control and *Tpr^flox/flox*;*Stra8-cre* mice. Scale bars, 50 μm. (G) Immunofluorescence staining of testicular sections from control and *Tpr^flox/flox*;*Stra8-cre* with antibodies against SYCP3 (red), a marker of spermatocyte. The nuclei were stained with Hoechst 33342 (blue). The magnified view of the boxed area is shown in the lower-left corner of the image. Scale bars, 50 μm.

