## [Peer Review File · EMBO Reports]

S100PBP interacts with the nucleoporin TPR and facilitates XY crossover formation in mice

Yufan Wu, Yang Li, Huan Zhang, Jingwei Ye, Ming Li, Jianteng Zhou, Xuefeng Xie, Hao Yin, Min Chen, Gang Yang, Suixing Fan, Baolu Shi, Hanwei Jiang, Qinghua Shi, and Hui Ma

Corresponding author(s): Hui Ma (clsmh@ustc.edu.cn), Qinghua Shi (qshi@ustc.edu.cn), Hanwei Jiang (hwj1209@ustc.edu.cn)

Review Timeline:

Submission Date:	21st Jun 24
Editorial Decision:	23rd Jul 24
Revision Received:	22nd Oct 24
Editorial Decision:	29th Nov 24
Revision Received:	26th Dec 24
Accepted:	28th Jan 25

Editor: Deniz Senyilmaz Tiebe

Transaction Report:

Dear Dr. Ma,

Thank you for transferring your research manuscript to our journal, which was now seen by three referees, whose reports are copied below.

Referees express interest in the proposed role of S100PBP in XY crossover formation in mice. However, they also raise significant concerns that need to be addressed to consider publication here.

I find the reports informed and constructive, and believe that addressing the concerns raised will significantly strengthen the manuscript. As the reports are below, and I think all points need to be addressed, I will not detail them here.

Given these positive recommendations, we would like to invite you to submit a revised manuscript. Please revise your manuscript with the understanding that the referee concerns (as in their reports) must be fully addressed and their suggestions taken on board. Please address all referee concerns in a complete point-by-point response. Acceptance of the manuscript will depend on a positive outcome of a second round of review. It is EMBO reports policy to allow a single round of major experimental revision only and acceptance or rejection of the manuscript will therefore depend on the completeness of your responses included in the next, final version of the manuscript.

We realize that it is difficult to revise to a specific deadline. In the interest of protecting the conceptual advance provided by the work, we recommend a revision within 3 months. Please discuss the revision progress ahead of this time with me if you require more time to complete the revisions, or if you have questions or comments regarding the revision (also by video chat).

1. A data availability section providing access to data deposited in public databases is missing (where applicable).
2. Your manuscript contains statistics and error bars based on $n=2$. Please use scatter plots in these cases.

You can submit the revision either as a Scientific Report or as a Research Article. For Scientific Reports, the revised manuscript can contain up to 5 main figures and 5 Expanded View figures, and it should not exceed 27000 characters. If the revision leads to a manuscript with more than 5 main figures it will be published as a Research Article. In this case the Results and Discussion section should be separate. If a Scientific Report is submitted, these sections have to be combined. This will help to shorten the manuscript text by eliminating some redundancy that is inevitable when discussing the same experiments twice. In either case, all materials and methods should be included in the main manuscript file.

4) a .docx formatted letter INCLUDING the reviewers' reports and your detailed point-by-point responses to their comments. As part of the EMBO publication's Transparent Editorial Process, EMBO reports publishes online a Review Process File (RPF) to accompany accepted manuscripts. This File will be published in conjunction with your paper and will include the referee reports, your point-by-point response and all pertinent correspondence relating to the manuscript.

<https://www.embopress.org/page/journal/14693178/authorguide#transparentprocess>

5) a complete author checklist, which you can download from our author guidelines

<https://www.embopress.org/page/journal/14693178/authorguide>. Please insert information in the checklist that is also reflected in the manuscript. The completed author checklist will also be part of the RPF.

6) Please note that all corresponding authors are required to supply an ORCID ID for their name upon submission of a revised manuscript (<<https://orcid.org/>>). Please find instructions on how to link your ORCID ID to your account in our manuscript tracking system in our Author guidelines

<<https://www.embopress.org/page/journal/14693178/authorguide#authorshipguidelines>>

7) Before submitting your revision, primary datasets produced in this study need to be deposited in an appropriate public database (see <https://www.embopress.org/page/journal/14693178/authorguide#datadeposition>). Please remember to provide a reviewer password if the datasets are not yet public. The accession numbers and database should be listed in a formal "Data Availability" section placed after Materials & Method (see also

<https://www.embopress.org/page/journal/14693178/authorguide#datadeposition>). Please note that the Data Availability Section is restricted to new primary data that are part of this study. * Note - All links should resolve to a page where the data can be accessed. *

Additional information on source data and instruction on how to label the files are available:

<https://www.embopress.org/page/journal/14693178/authorguide#sourcedata>

9) Our journal encourages inclusion of *data citations in the reference list* to directly cite datasets that were re-used and obtained from public databases. Data citations in the article text are distinct from normal bibliographical citations and should directly link to the database records from which the data can be accessed. In the main text, data citations are formatted as follows: "Data ref: Smith et al, 2001" or "Data ref: NCBI Sequence Read Archive PRJNA342805, 2017". In the Reference list, data citations must be labeled with "[DATASET]". A data reference must provide the database name, accession number/identifiers and a resolvable link to the landing page from which the data can be accessed at the end of the reference. Further instructions are available at <http://www.embopress.org/page/journal/14693178/authorguide#referencesformat>

10) Regarding data quantification (see Figure Legends:

<https://www.embopress.org/page/journal/14693178/authorguide#figureformat>)

12) Please also note our reference format:

13) All Materials and Methods need to be described in the main text. We would encourage you to use 'Structured Methods', our new Methods format. According to this format, the Methods section should include a Reagents and Tools Table (listing key reagents, experimental models, software and relevant equipment and including their sources and relevant identifiers) followed by a Methods and Protocols section in which we encourage the authors to describe their methods using a step-by-step protocol format with bullet points, to facilitate the adoption of the methodologies across labs. More information on how to adhere to this format as well as downloadable templates (.doc or .xls) for the Reagents and Tools Table can be found in our author guidelines: <https://www.embopress.org/page/journal/14693178/authorguide#manuscriptpreparation>.

An example of a Method paper with Structured Methods can be found here: <https://www.embopress.org/doi/full/10.1038/s44320-024-00037-6#sec-4>

I look forward to seeing a revised version of your manuscript when it is ready. Please let me know if you have questions or comments regarding the revision.

Kind regards,

Deniz Senyilmaz Tiebe

Deniz Senyilmaz Tiebe, PhD
Scientific Editor
EMBO Reports

Referee #1:

Wu et al. identify S100PBP as being important for normal spermatogenesis in mice (but not for oogenesis). They show that S100PBP^{-/-} mutants are specifically deficient in crossover in the pseudoautosomal region shared by the X and Y chromosomes; these mutants also show a modest elevation in autosomal crossover (as assayed by scoring Mlh1/Mlh3 foci in the PAR). S100PBP is shown to localize to the nuclear periphery of spermatocytes through an interaction with the nuclear pore complex protein TPR. Experiments to characterize this interaction further are presented but are problematic.

Overall, the work in this manuscript is well done and supports the conclusion. S100PBP seems to have a new and unique role in mammalian meiosis, and this manuscript is appropriate for EMBO Reports, once revised.

The following concerns are presented in order of appearance in the manuscript; particularly important concerns are marked with an asterisk (*)

1. It would be very helpful if proteins, when first introduced, are briefly described. See line 85 for a good example of how to do this.
2. Lines 177 and following. It appears that 80% of spermatocytes have univalent sex chromosomes at metaphase I, but only 70% of spermatocytes lack an Mlh1 focus in the PAR. Does this mean that an Mlh1 focus does not guarantee a crossover in the mutant?
3. *Line 200 and following. In the two examples given in Figure EV2, it appears that the X and Y chromosomes are colocalized even though they are not synapsed. It would be useful to quantify the fraction of nuclei where the sex chromosomes are colocalized but not synapsed, the fraction where they are synapsed, and the fraction where they are not colocalized.
4. Line 209. The mutant is not a deletion of S100pbp, it is a frameshift mutation. I suggest "S100pbp mutation confers.."
5. Line 248 and following. While TEX11 and M1AP foci in the PAR are reduced, they are reduced only modestly, and not to levels sufficient to explain the crossover defect. This should be discussed.
6. Line 256 and following. It's important to consider that the autosomal effect of S100pbp mutation may be indirect-for example, that the absence of a crossover in the PAR is sensed and either progression is delayed or recombination activity is increased, leading to additional COs on the autosomes.
7. *Line 357 and following. The pulldown experiments using protein deletions are of poor quality and should probably be dropped. The concern is that with such widely differing levels of the GFP-S100PBP protein fusions, it will be impossible to determine where the defect lies. For example, ΔC is scored as still interacting with TPR, but the levels of this protein are very high and it looks as if the efficiency of pulldown is much lower than with the WT protein. Quantitative analysis may help, but when the levels of fusion protein expression vary so widely, the concern is that the more weakly-expressed fusion proteins may still interact with TPR, but the pulldown levels are below detection. At a very minimum, reciprocal pulldowns (pull down TPR, score for GFP-S100PBP) should be done, and all require careful quantification.
8. *Line 362 and following. If it cannot be shown that S100PBP- $\Delta 27-85$ aa protein is present and expressed at wild-type levels,

then the phenotype of this mutant cannot be interpreted. This section should be dropped.

Even if the data in Figure 7 and Figures EV9 and EV10 are removed from the manuscript, it will be of sufficient interest to justify publication at EMBO Reports.

Referee #2:

This research group has made very significant contributions to the understanding of mechanisms related to chromosome segregation, mammalian meiosis and gametogenesis.

The study does contain multiple interesting findings and not a single key finding. It is of great significance for the field of spermatogenesis and meiotic recombination, and of general interest for the molecular biology community.

Here the authors have identified a novel modulator of mammalian meiotic recombination which facilitates sex chromosomes recombination in males but also constrains autosomal recombination in both sexes. Interestingly, this protein appears to act through an interaction via its N-terminus with a component of the nuclear pore complex, a structure not suspected to play a direct role in meiotic recombination.

Overall, I consider the manuscript well written, the reported findings are novel and quite interesting, well presented (figures are clear) and convincing. The conclusions are well supported by the data and are appropriately discussed in the context of earlier literature. Moreover, the finding that the Nuclear Pore Complex is required for normal meiotic recombination opens new doors for interesting research.

Here is a more detailed list of comments, mostly comments of minor importance.

Major comments

Line 362: "suitable antibody". Given the high number of antibodies that the group has produced using ABclonal services in this study (including antibodies already commercially available), one can wonder why the group did not create another antibody anti-S100PBP that could recognize the truncated version S100PBPdelta177 protein, or made a mouse with a tagged version of S100PBPdelta177, in order to fully prove that the N-term of S100PBP is the key domain explaining the phenotype. However, this weakness is well acknowledged and discussed in the manuscript, and I do not consider the long work associated with such additional experiment crucial or worth doing at this point.

Line 413: "data not shown". The increase of autosomal crossovers (CO) number is a very interesting observation. Not much is known about mutants increasing CO number so this effect is hard to explain, however the strong and well-known correlation between crossover number and axis length makes a putative increase of axis length a very good candidate to explain the observed increase in CO numbers. A subtle increase of axis length wouldn't be obvious to detect by eye and therefore should be precisely measured. I think the authors can easily perform this additional experiment in order to be able to conclude that this increase of CO number is indeed not caused by an increase in axis length in the mutant.

Minor comments

Line 23: The manuscript contains of mixture of "crossovers" and "crossing-over". The authors should choose one term for consistency.

Line 47: "crossovers are the key event of meiosis". It is a detail but meiosis is a long and multistep process, especially mammalian meiosis, and I don't see how one could pinpoint "the one key event" of meiosis.

Line 50: the acronym DSB must be explained here (double-strand breaks).

Line 64: "adjacent crossovers are widely distributed". This is unclear and should be rephrased, maybe simply removing "adjacent" is good enough.

Line 66: "the number of crossovers is maintained at a constant level". This is unclear. Number of CO maintained constant compared to what? From cell to cell? From male to female meiocytes? Across species? With more or less DSBs? Or is it the ratio of crossovers/DSBs that is maintained constant?

Line 97: "Although the functions of many genes in spermatogenesis have been resolved, others remain enigmatic." This is very much a detail but this sentence doesn't add anything. Nobody expects that among the 2300 genes specifically expressed in the testes, all of them or none them have a well-known function.

Line 127: Unclear phrasing, maybe remove « the meiosis initiation of ».

Line 159: "Kauppi et al., 2011; Ma et al., 2022". It is a detail but I find this to be a strange choice of references for the misalignment of univalents. Achiasmatic chromosomes failing to align in MMI was discovered before 2011.

Line 168: "in wild-type mice" or "in control mice" must be specified.

Line 171-172: "Likely"? Few lines above the authors were able to precisely quantify the number of spermatocytes with univalent X and Y, to finally conclude here that these univalent are only "likely" to be X and Y? This is confusing and prompts to reader to question the ability of the authors to precisely identify the sex chromosomes in metaphase I. It should be rephrased.

Line 214: alterations

Line 336: Precise referencing to Fig 6A and 6B must be made (not just Fig 6).

Line 340, Fig 6: It is disturbing for the reader to see that deltaM migrates faster than deltaC and that deltaN1 migrates as fast as deltaC, given the differences in number of deleted amino acids. Simply asking for a double check or for a comment on this.

Line 475: "milder". Instead of stating in the result section that the phenotype of the interaction mutant is simply "similar" to the null, and waiting for the discussion section to confess that the phenotype is actually milder, I think it is important to state that the phenotype is milder in the result section as well.

Line 803, Fig2 legend, and line 974, Fig EV10E legend: "Frequencies of nuclei with an MLH1 focus". Which stage of cells exactly? Early, mid, late pachytene, or all pachytene substages combined?

Line 811, Fig3 A,D,G legend. Why are bottom-left inlets empty (black) on the images? Is there a H1t staining here? If yes this must be specified in the legend.

Line 817, Fig3 legend, same as Line 901, Fig EV3 legend: "The mean number ". The figure is not just a mean number, it is a scatter plot graph showing the total foci number with the mean number indicated by a black bar.

Fig 3B: A tiny white bar is lost above the data points in the 4th row of Fig 3B.

Referee #3:

In this study, by generating and characterizing mutant mouse models, Wu et al. show that S100PBP is essential for fertility in male because XY chromosomes frequently fail to align correctly at meiotic metaphase I, due to defective recombination. On the contrary, the number of crossovers was increased in oocytes and on autosomes in males. They identify the nuclear pore component TPR as a S100PBP interactor, and show that S100PBP localizes to nuclear pore during early meiotic prophase in a Tpr-dependent manner.

Altogether, this study points to an indirect role for S100PBP. It deserves the credit of pinpointing S100PBP as a putative nuclear pore-associated factor involved in regulating nuclear events during meiotic prophase. However, one weakness of this report is that it does not give any insight into what might be the role of S100PBP at the nuclear pores. Also, downstream, the observed phenotype may result from the deregulation of one or several nuclear factors, which are not identified and cannot be told apart. This work brings some novelty on a gene previously proposed to be involved in tumorigenesis, and might open the way for studying the role of nuclear pores in this context, but as of now the results are rather preliminary and would not allow a broad audience reader to get insights into the involved biological processes. My current view is that these data should be reported in a more specialized journal than EMBO reports.

In general, the experiments aiming at characterizing the meiotic defects of the mutants are well carried out, with appropriate controls. The interaction with TPR and the nuclear pore localization are rather convincing, although the characterization of the nuclear pore localization of S100PBP is rudimentary and there is no attempt to understand its function. I have several comments that should be responded to.

General comments:

Because the main conclusions are clear but do not allow to deduce mechanisms, the interpretations and the discussion should be more focused and shortened. The localization of S100PBP suggests an indirect function, as discussed (lines 439-468). Because a likely hypothesis is that one or more unidentified nuclear factors are somehow deregulated, the first part of the discussion (lines 388-438) might come after and should be shortened to give a shorter overview of the possible involved factors, with the possibility that several independent factors are involved. Please mention that a relatively weak change in the abundance of some factors might alter recombination (e.g., Rnf212+/-, Reynolds et al, nature genetics, 2013).

In theory, one single possible mechanism explaining an opposite trend on autosomes (more CO) and PAR (less frequent CO) could be by lowering CO interference and assurance. This might result (given certain parameters) in more CO on autosomes and a reduced CO assurance exacerbated on XY. A better characterization of MLH1/3 focus number (count autosomes with 2, 3 COs per nucleus without pooling 2 and 3 foci-containing autosomes) and distribution (distances between MLH1/3 foci on autosomes with 2 or 3 foci) on autosomes might improve the characterization of this aspect of the phenotype.

One limitation of the study resides in the uncertainty about the alleles. Indeed, they authors acknowledge (line 360) and discuss (e.g., lines 469-480) that they cannot know whether the D177bp allele produces any stable protein at all, because the anti-S100PBP antibody directed against aa 19-33 would not detect it. For the same reason, the frameshift allele may or may not produce a truncated protein initiating at an internal ATG (e.g., methionine 60). Therefore, because both alleles may formally either be null or produce a truncated protein lacking the N-ter region, the observed phenotype of either allele cannot be attributed with certainty to either a complete loss of function or a truncation of the N-terminal region (possibly with reduced expression). Although this might not impact strongly the conclusion of the study, this should be mentioned in the manuscript, and the lines 469-484 of the discussion should be shortened or replaced by 1-2 sentences in the Results section.

Specific comments:

Line 151: Is there any difference between +/+ and heterozygous mice, especially for counts of univalent XY and foci? Some heterozygous mutants were previously shown to result in altered crossover numbers (e.g., *Rnf212+/-*). Please specify, which genotypes are included as controls.

154: as bivalents (plural).

165: fig 1H, n= ?

Lines 175-195: The PAR with MLH1/3 foci counted for XY bivalents, or for all XY including separated ones? Because the proportions of bivalent XY (Fig. 1I) and of XY with a MLH1/3 focus (Fig. 2B,F) are similar, I wonder whether virtually every bivalent XY has a MLH1/3 focus in *S100pbp-/-* spermatocytes. Please clarify and indicate whether there are MLH1/3 foci on PAR of separated XY.

183: Please indicate the age of the mice examined for MLH1 (and other) counts. Spermatocytes from adult and juvenile mice have been reported to have different MLH1-MLH3 (and RPA2) counts (Zelazowski et al., Cell, 2017), therefore age differences might alter the results.

Lines 220-223: The counts of nuclei with RPA2 and RAD51 are shown only at the PAR of spermatocytes "with XY touching" on Fig. EV3 E-H. However, it is written in the main text that these counts are "comparable" for spermatocytes "with XY separated or touching". The counts of PAR of separated XY with RPA2 and RAD51 foci must also be shown. The criteria used to determine whether foci overlap the PAR must be indicated (e.g., PAR FISH, or specific distance measured from X or Y axis end).

286: To identify S100PBP interactors, testicular lysates should have been compared between age-matched juvenile animals with similar cell composition: 10-week-old *S100pbp-/-* testes are devoid of post-MI cells whereas WT testes are filled with haploid cells. Please show the list of peptides found in control IPs (IgG and *S100pbp-/-*) on Table S2.

412: the data showing that there is no change in chromosome length (SC in pachytene stage?) should be shown.

Here are our point-by-point responses to reviews:

We would like to thank you for taking the time and effort necessary to review our manuscript. We sincerely appreciate all your valuable comments and suggestions, which have greatly helped us to improve the quality of the manuscript.

Referee #1:

Wu et al. identify S100PBP as being important for normal spermatogenesis in mice (but not for oogenesis). They show that S100PBP^{-/-} mutants are specifically deficient in crossover in the pseudoautosomal region shared by the X and Y chromosomes; these mutants also show a modest elevation in autosomal crossover (as assayed by scoring Mlh1/Mlh3 foci in the PAR). S100PBP is shown to localize to the nuclear periphery of spermatocytes through an interaction with the nuclear pore complex protein TPR. Experiments to characterize this interaction further are presented but are problematic.

Overall, the work in this manuscript is well done and supports the conclusion. S100PBP seems to have a new and unique role in mammalian meiosis, and this manuscript is appropriate for EMBO Reports, once revised.

The following concerns are presented in order of appearance in the manuscript; particularly important concerns are marked with an asterisk (*)

1. It would be very helpful if proteins, when first introduced, are briefly described. See line 85 for a good example of how to do this.

Response: We thank you for the suggestion. We have added descriptions where necessary; please refer to lines 34, 63, and 88, etc.

2. Lines 177 and following. It appears that 80% of spermatocytes have univalent sex chromosomes at metaphase I, but only 70% of spermatocytes lack an Mlh1 focus in the PAR. Does this mean that an Mlh1 focus does not guarantee a crossover in the mutant?

Response: Thank you for the comment. Taking into account your concerns, as well as those raised by Reviewer #2 (comment 17) and Reviewer #3 (comment 7), we have re-analyzed the MLH1 foci at the mid-pachytene; focusing on this stage because MLH1 focus is absent at the PAR in most wild-type (WT) spermatocytes, though XY PARs remain touching, during late pachytene, which could affect the interpretation of the results.

At metaphase I, the average percentage of nuclei with univalent XY is 12.86% in controls and 74.83% in the mutants, with a difference of 61.97% (Fig 1I). At the mid-pachytene stage, the average percentage of nuclei lacking an MLH1 focus at the PAR is 8.12% in controls and 67.35% in the mutants, with a difference of 59.23% (Fig 2B). The differences are comparable between the pachytene stage (59.23%) and the metaphase I (61.97%). Therefore, we believe that the presence of an MLH1 focus at the mid-pachytene stage would guarantee a crossover between the X and Y chromosomes at the metaphase stage in the mutants.

3. *Line 200 and following. In the two examples given in Figure EV2, it appears that the X and Y chromosomes are colocalized even though they are not synapsed. It would be useful to quantify the fraction of nuclei where the sex chromosomes are colocalized but not synapsed, the fraction where they are synapsed, and the fraction where they are not colocalized.

Response: Thank you for the comment and suggestion. In the initial submission, we quantified the fraction of nuclei with untouching XY PARs (which should not be described as “XY untouching”), because the signal of Six6OS1 (which indicates synapsis) is absent from the PARs in most WT spermatocytes during late pachytene, although the XY PARs remain touched, which could interfere with the interpretation of the results. In the examples provided in Fig. EV2 of the initial submission (Fig. EV1C in the revised manuscript), we aimed to show that the PARs of the XY chromosomes are untouching and not synapsed.

As you point out, it would be more precise to differentiate the “XY synapsed”, “XY unsynapsed but still touching at the PARs”, and “XY PARs untouching”. Following your suggestion, we re-quantified the fraction of nuclei showing XY asynapsis and the fraction of nuclei showing untouching XY PARs at the early and mid-pachytene stage (Fig EV1 D and E). Specifically, among the 69 early pachytene spermatocytes analyzed for *S100pbp*^{-/-} mice, 23 exhibited XY asynapsis, and the PARs of X and Y are untouching in all these 23 nuclei. Among the 66 mid-pachytene mutant spermatocytes, 46 exhibited XY asynapsis, with only 3 of these 46 nuclei having touching XY PARs. Thus, we infer that XY chromosomes failing to synapse but remain touching at the PAR is not a frequent occurrence in the mutant.

To avoid any confusion, we have revised the manuscript to modify “XY separation/untouching” as “untouching XY PARs” and replaced the representative images in Fig. EV1C. The frequencies of the XY asynapsis and untouching XY PARs are shown in Fig. EV1 D and E.

4. Line 209. The mutant is not a deletion of S100pbp, it is a frameshift mutation. I suggest "S100pbp mutation confers.."

Response: Thank you for the comment and suggestion. We have made changes accordingly and at places where appropriate.

5. Line 248 and following. While TEX11 and M1AP foci in the PAR are reduced, they are reduced only modestly, and not to levels sufficient to explain the crossover defect. This should be discussed.

Response: Thank you for the comment and suggestion. We agree with you that the reduced frequencies of TEX11 and M1AP foci at the PAR are not sufficient to fully explain

the crossover defect observed in *S100pbp* knockouts, but we think it could partially account for the XY asynapsis occurring from the early pachytene to the mid-pachytene stage. Because the reduction of the frequency of cells harboring TEX11/M1AP foci at the PAR with touching XY PARs in early pachytene nuclei (69.58%/69.02% in *S100pbp* knockouts versus 95.24%/97.44% in the controls) parallels with the increase in the frequencies of XY separation in *S100pbp* knockouts from the early to the mid pachytene stage (controls: 6.55% and 13.76% in the early and mid pachytene stage, respectively; *S100pbp*^{-/-}: 35.53% and 65.32% in the early and mid pachytene stage, respectively). We have included this in the discussion (line 276 and lines 460-468) in the revised manuscript.

6. Line 256 and following. It's important to consider that the autosomal effect of *S100pbp* mutation may be indirect-for example, that the absence of a crossover in the PAR is sensed and either progression is delayed or recombination activity is increased, leading to additional COs on the autosomes.

Response: Thank you for the comment and suggestion. We agree that the autosomal effects of the *S100pbp* mutation may be indirect, and we have included this in the Discussion section of the revised manuscript (lines 407-429). The assumptions you raised are also reasonable; however, we currently do not have sufficient evidence to support these claims.

7. *Line 357 and following. The pulldown experiments using protein deletions are of poor quality and should probably be dropped. The concern is that with such widely differing levels of the GFP-*S100PBP* protein fusions, it will be impossible to determine where the defect lies. For example, ΔC is scored as still interacting with TPR, but the levels of this protein are very high and it looks as if the efficiency of pulldown is much lower than with the WT protein. Quantitative analysis may help, but when the levels of fusion protein expression vary so widely, the concern is that the more weakly-expressed fusion proteins may still interact with TPR, but the pulldown levels are below detection. At a very minimum, reciprocal pulldowns (pull down TPR, score for GFP-*S100PBP*) should be done, and all require careful quantification.

Response: Thank you for the comment and suggestion. We have repeated the pulldown experiments using protein deletions, ensuring that the expression levels of the GFP-fusion proteins are similar. The efficiencies of the GFP-fusion proteins pulled-down by the anti-GFP antibodies are also similar, except for GFP-*S100PBP*- ΔM which remains at a markedly lower level in the GFP-pulldown lysates than the other fusion proteins. Despite this, the low amount of GFP-*S100PBP*- ΔM is still able to pull down TPR, whereas GFP-*S100PBP*- ΔN , GFP-*S100PBP*- $\Delta N1$, GFP-*S100PBP*- $\Delta N2$, and GFP-*S100PBP*- ΔIDR do not. Collectively, the results are in congruent with our previous conclusion that the *S100PBP* N-terminus is essential for interacting with TPR. The new results are now shown in Fig. 6B.

We also performed the reciprocal pulldowns by using the GFP-antibody conjugated beads to pull down GFP-TPR, and scored for FLAG-S100PBP. The results show that FLAG antibody successfully detected the FLAG-S100PBP in the GFP-pulled down cell lysates, further confirming the interaction between S100PBP and TPR. The new results are now shown in Fig. EV4B.

8. *Line 362 and following. If it cannot be shown that S100PBP- Δ 27-85aa protein is present and expressed at wild-type levels, then the phenotype of this mutant cannot be interpreted. This section should be dropped.

Even if the data in Figure 7 and Figures EV9 and EV10 are removed from the manuscript, it will be of sufficient interest to justify publication at EMBO Reports.

Response: Thank you for your thoughtful comment. As we discussed in the second-to-last paragraph of the Discussion (lines 475-488), we acknowledge that it cannot definitively attribute the observed phenotype to a complete loss of function or the deletion of the N-terminal domain; however, the similar phenotypes observed in *S100pbp* ^{Δ 177bp/ Δ 177bp} mice and *S100pbp*^{-/-} mice would lend support to either the hypothesis that the N-terminal IDR domain is specifically required for stabilizing the S100PBP protein in testes or the hypothesis that the N-terminus interaction with TPR mediates the biological function of S100PBP by anchoring it to the nuclear pores (as inferred from the mislocalization of S100PBP in *Tpr*-deficient spermatocytes). These *in vivo* findings underscore the importance of the N-terminus and indicate potential molecular mechanisms for S100PBP function, providing directions for future studies. Therefore, we would prefer to retain these results.

Referee #2:

This research group has made very significant contributions to the understanding of mechanisms related to chromosome segregation, mammalian meiosis and gametogenesis.

The study does contain multiple interesting findings and not a single key finding. It is of great significance for the field of spermatogenesis and meiotic recombination, and of general interest for the molecular biology community.

Here the authors have identified a novel modulator of mammalian meiotic recombination which facilitates sex chromosomes recombination in males but also constrains autosomal recombination in both sexes. Interestingly, this protein appears to act through an interaction via its N-terminus with a component of the nuclear pore complex, a structure not suspected to play a direct role in meiotic recombination.

Overall, I consider the manuscript well written, the reported findings are novel and quite interesting, well presented (figures are clear) and convincing. The conclusions are well supported by the data and are appropriately discussed in the context of earlier literature.

Moreover, the finding that the Nuclear Pore Complex is required for normal meiotic recombination opens new doors for interesting research.

Here is a more detailed list of comments, mostly comments of minor importance.

Major comments

1. Line 362: "suitable antibody". Given the high number of antibodies that the group has produced using ABclonal services in this study (including antibodies already commercially available), one can wonder why the group did not create another antibody anti-S100PBP that could recognize the truncated version S100PBPdelta177 protein, or made a mouse with a tagged version of S100PBPdelta177, in order to fully prove that the N-term of S100PBP is the key domain explaining the phenotype. However, this weakness is well acknowledged and discussed in the manuscript, and I do not consider the long work associated with such additional experiment crucial or worth doing at this point.

Response: Thank you for your understanding. We attempted to generate antibodies that could bind to an epitope of the N-terminus of the S100PBP protein; however, unfortunately, the antibody did not work well.

2. Line 413: "data not shown". The increase of autosomal crossovers (CO) number is a very interesting observation. Not much is known about mutants increasing CO number so this effect is hard to explain, however the strong and well-known correlation between crossover number and axis length makes a putative increase of axis length a very good candidate to explain the observed increase in CO numbers. A subtle increase of axis length wouldn't be obvious to detect by eye and therefore should be precisely measured. I think the authors can easily perform this additional experiment in order to be able to conclude that this increase of CO number is indeed not caused by an increase in axis length in the mutant.

Response: Thank you for your comment and constructive suggestion. Following your advice, we have carefully measured the autosomal axis lengths of the mid-pachytene *S100pbp*^{-/-} spermatocytes and compared them to those in the control cells. As shown in **Appendix Fig. S3B**, the average axis length of the autosomes is slightly greater in *S100pbp*^{-/-} spermatocytes than that in the controls; however, this difference does not reach statistical significance.

Minor comments

3. Line 23: The manuscript contains of mixture of "crossovers" and "crossing-over". The authors should choose one term for consistency.

Response: Thank you for your careful reading and suggestion. We were fully aware of the importance of consistency in terminology. We opted to use "crossover" to refer to the final outcome of recombination and "crossing-over" to describe the recombination process

that leads to crossovers. This distinction aligns with common usage in the literature. Several research groups have used both “crossover” and “crossing-over/crossing over” in their papers, as seen in (Bondarieva et al, Nat Commun, 2020), (Condezo et al, PNAS, 2024), (Rao et al, Science, 2017), and (Xu et al, PNAS, 2023), among others. We appreciate your suggestion and have chosen to use “crossover” consistently throughout the manuscript.

References:

Bondarieva A, Raveendran K, Telychko V, et al., Proline-rich protein PRR19 functions with cyclin-like CNTD1 to promote meiotic crossing over in mouse. Nat Commun. 2020 Jun 18;11(1):3101. doi: 10.1038/s41467-020-16885-3.

Condezo, Sainz-Urruela, Gomez-H et al., RNF212B E3 ligase is essential for crossover designation and maturation during male and female meiosis in the mouse, Proc Natl Acad Sci U S A. 2024 Jun 18;121(25):e2320995121. doi: 10.1073/pnas.2320995121.

Rao H B D P, Qiao H, Bhatt S K, et al, A SUMO-ubiquitin relay recruits proteasomes to chromosome axes to regulate meiotic recombination. Science. 2017 Jan 27;355(6323):403-407. doi: 10.1126/science.aaf6407.

Xu J, Li T, Kim S, Boekhout M, Keeney S, Essential roles of the ANKRD31-REC114 interaction in meiotic recombination and mouse spermatogenesis. Proc Natl Acad Sci U S A. 2023 Nov 21;120(47):e2310951120. doi: 10.1073/pnas.2310951120.

4. Line 47: "crossovers are the key event of meiosis". It is a detail but meiosis is a long and multistep process, especially mammalian meiosis, and I don't see how one could pinpoint "the one key event" of meiosis.

Response: Thank you for your careful reading and suggestion. We have re-worded the sentence to “... are one of the key events...” in the revised manuscript (line 48).

5. Line 50: the acronym DSB must be explained here (double-strand breaks).

Response: Thank you for your careful reading and suggestion. We have made modifications accordingly in the revised manuscript (line 51).

6. Line 64: "adjacent crossovers are widely distributed". This is unclear and should be rephrased, maybe simply removing "adjacent" is good enough.

Response: Thank you for your careful reading and suggestion. We have removed “adjacent” (line 66).

7. Line 66: "the number of crossovers is maintained at a constant level". This is unclear.

Number of CO maintained constant compared to what? From cell to cell? From male to female meiocytes? Across species? With more or less DSBs? Or is it the ratio of crossovers/DSBs that is maintained constant?

Response: Thank you for your careful reading and suggestion. We completely agree with your point. In the revised manuscript, we have reworded the sentence to "the average number of crossovers per cell is well-controlled (at around 24 per spermatocytes in wild-type mice)" (lines 68-70).

8. Line 97: "Although the functions of many genes in spermatogenesis have been resolved, others remain enigmatic." This is very much a detail but this sentence doesn't add anything. Nobody expects that among the 2300 genes specifically expressed in the testes, all of them or none them have a well-known function.

Response: Thank you for your careful reading and comment. We have removed this sentence from the revised manuscript.

9. Line 127: Unclear phrasing, maybe remove « the meiosis initiation of ».

Response: Thank you for your careful reading and suggestion. We have removed this from the manuscript.

10. Line 159: "Kauppi et al., 2011; Ma et al., 2022". It is a detail but I find this to be a strange choice of references for the misalignment of univalents. Achiasmatic chromosomes failing to align in MMI was discovered before 2011.

Response: Thank you for your careful reading and suggestion. We agree with you and have corrected the citation (line 156).

11. Line 168: "in wild-type mice" or "in control mice" must be specified.

Response: Thank you for your careful reading and suggestion. We agree with you and have corrected this and at places where necessary.

12. Line 171-172: "Likely"? Few lines above the authors were able to precisely quantify the number of spermatocytes with univalent X and Y, to finally conclude here that these univalent are only "likely" to be X and Y? This is confusing and prompts to reader to question the ability of the authors to precisely identify the sex chromosomes in metaphase I. It should be rephrased.

Response: Thank you for your careful reading and suggestion. We used "likely" because we did not directly demonstrate the unaligned chromosomes are the sex chromosomes

through additional experiments (such as fluorescence *in situ* hybridization). However, based on the high incidence of univalent XY chromosomes in the mutant MMI cells, we inferred that the unaligned chromosomes observed in *S100pbp*^{-/-} MMI spermatocytes are the univalent XY chromosomes. We agree with your suggestion and have reworded this sentence as “based on these results, we believe that the unaligned chromosomes observed in *S100pbp*^{-/-} MMI spermatocytes are the univalent XY chromosomes.”

13. Line 214: alterations

Response: Thank you for your careful reading. We have corrected this.

14. Line 336: Precise referencing to Fig 6A and 6B must be made (not just Fig 6).

Response: Thank you for your careful reading. We have corrected this.

15. Line 340, Fig 6: It is disturbing for the reader to see that deltaM migrates faster than deltaC and that deltaN1 migrates as fast as deltaC, given the differences in number of deleted amino acids. Simply asking for a double check or for a comment on this.

Response: Thank you for your careful reading. These experiments have been performed at least three independent times, and the results have consistently shown the same pattern. We also noticed that the detected band size of wild-type mouse S100PBP protein in both the testicular lysates and cultured cell lines is larger than the molecular weight calculated from the protein sequences (44.5 kD), S100PBP-ΔM (Δ156-269 aa) migrates faster than S100PBP-ΔC (270-396 aa), S100PBP-ΔN1 (Δ2-50 aa) and S100PBP-ΔIDR (Δ27-94 aa) migrate at a similar rate as S100PBP-ΔC (270-396 aa). These observations may suggest potential post-translational modifications in the N-terminal and Middle regions of S100PBP protein.

16. Line 475: "milder". Instead of stating in the result section that the phenotype of the interaction mutant is simply "similar" to the null, and waiting for the discussion section to confess that the phenotype is actually milder, I think it is important to state that the phenotype is milder in the result section as well.

Response: Thank you for your careful reading and constructive suggestion. We have stated in the result section that the phenotype is milder (line 383).

17. Line 803, Fig2 legend, and line 974, Fig EV10E legend: "Frequencies of nuclei with an MLH1 focus". Which stage of cells exactly? Early, mid, late pachytene, or all pachytene substages combined?

Response: Thank you for your careful reading and constructive suggestion. Following the

recommendations from reviewer #1 (comment 2) and reviewer #3 (comment 7), we re-quantified the number of MLH1 foci focusing on the mid-pachytene stage. The stage of cells is now stated in the results section (line 176 and line 390) and the corresponding figure legends (Fig. 2 and Fig. EV5).

18. Line 811, Fig3 A,D,G legend. Why are bottom-left inlets empty (black) on the images? Is there a H1t staining here? If yes this must be specified in the legend.

Response: Thank you for your careful reading. We apologize for the oversight. The bottom-left insets are indeed H1t staining, which is used to differentiate between the early pachytene stage (H1t-negative) and other pachytene cells (H1t-positive). This clarification has been added to the corresponding figure legend (Fig. 3).

19. Line 817, Fig3 legend, same as Line 901, Fig EV3 legend: "The mean number ". The figure is not just a mean number, it is a scatter plot graph showing the total foci number with the mean number indicated by a black bar.

Response: Thank you for pointing it out. We agree with you. We have corrected the figure legends (Fig. 3 and Fig. EV2).

20. Fig 3B: A tiny white bar is lost above the data points in the 4th row of Fig 3B.

Response: Thank you for your careful reading. We have removed the white bar.

Referee #3:

In this study, by generating and characterizing mutant mouse models, Wu et al. show that S100PBP is essential for fertility in male because XY chromosomes frequently fail to align correctly at meiotic metaphase I, due to defective recombination. On the contrary, the number of crossovers was increased in oocytes and on autosomes in males. They identify the nuclear pore component TPR as a S100PBP interactor, and show that S100PBP localizes to nuclear pore during early meiotic prophase in a Tpr-dependent manner.

Altogether, this study points to an indirect role for S100PBP. It deserves the credit of pinpointing S100PBP as a putative nuclear pore-associated factor involved in regulating nuclear events during meiotic prophase. However, one weakness of this report is that it does not give any insight into what might be the role of S100PBP at the nuclear pores. Also, downstream, the observed phenotype may result from the deregulation of one or several nuclear factors, which are not identified and cannot be told apart.

This work brings some novelty on a gene previously proposed to be involved in tumorigenesis, and might open the way for studying the role of nuclear pores in this

context, but as of now the results are rather preliminary and would not allow a broad audience reader to get insights into the involved biological processes. My current view is that these data should be reported in a more specialized journal than EMBO reports.

In general, the experiments aiming at characterizing the meiotic defects of the mutants are well carried out, with appropriate controls. The interaction with TPR and the nuclear pore localization are rather convincing, although the characterization of the nuclear pore localization of S100PBP is rudimentary and there is no attempt to understand its function. I have several comments that should be responded to.

General comments:

1. Because the main conclusions are clear but do not allow to deduce mechanisms, the interpretations and the discussion should be more focused and shortened. The localization of S100PBP suggests an indirect function, as discussed (lines 439-468). Because a likely hypothesis is that one or more unidentified nuclear factors are somehow deregulated, the first part of the discussion (lines 388-438) might come after and should be shortened to give a shorter overview of the possible involved factors, with the possibility that several independent factors are involved. Please mention that a relatively weak change in the abundance of some factors might alter recombination (e.g., Rnf212+/-, Reynolds et al, nature genetics, 2013).

Response: Thank you for your insightful suggestions and comments. We agree with your suggestions and have made the corresponding changes in the revised manuscript. These adjustments include adding information about the abundance of certain factors affecting recombination, swapping the second and third paragraphs, and rewriting parts of the discussion to make it more focused and concise. Please find in the revised manuscript (lines 407-488).

2. In theory, one single possible mechanism explaining an opposite trend on autosomes (more CO) and PAR (less frequent CO) could be by lowering CO interference and assurance. This might result (given certain parameters) in more CO on autosomes and a reduced CO assurance exacerbated on XY. A better characterization of MLH1/3 focus number (count autosomes with 2, 3 COs per nucleus without pooling 2 and 3 foci-containing autosomes) and distribution (distances between MLH1/3 foci on autosomes with 2 or 3 foci) on autosomes might improve the characterization of this aspect of the phenotype.

Response: Thank you for your insightful suggestions and comments. As you suggested, we have further analyzed the autosomes with 2, 3 COs per nucleus (the new result is shown in Fig. 2D, 2H, 4D and Fig. EV5F) and the distances between adjacent MLH1/3 foci on autosomes with 2 or 3 foci (Appendix Fig. S3A).

3. One limitation of the study resides in the uncertainty about the alleles. Indeed, they authors acknowledge (line 360) and discuss (e.g., lines 469-480) that they cannot know whether the D177bp allele produces any stable protein at all, because the anti-S100PBP antibody directed against aa 19-33 would not detect it. For the same reason, the frameshift allele may or may not produce a truncated protein initiating at an internal ATG (e.g., methionine 60). Therefore, because both alleles may formally either be null or produce a truncated protein lacking the N-ter region, the observed phenotype of either allele cannot be attributed with certainty to either a complete loss of function or a truncation of the N-terminal region (possibly with reduced expression). Although this might not impact strongly the conclusion of the study, this should be mentioned in the manuscript, and the lines 469-484 of the discussion should be shortened or replaced by 1-2 sentences in the Results section.

Response: Thank you for your insightful suggestion and comments. We agree with you. We have added these possibilities in the revised manuscript and shortened the part of the discussion you mentioned (lines 475-488).

Specific comments:

4. Line 151: Is there any difference between +/+ and heterozygous mice, especially for counts of univalent XY and foci? Some heterozygous mutants were previously shown to result in altered crossover numbers (e.g., Rnf212+/-). Please specify, which genotypes are included as controls.

Response: Thank you for your insightful suggestions and comments. We did not observe any obvious defects in fertility and spermatogenesis of *S100pbp*^{+/-} mice. Specifically, the testis weight, sperm count, the number of autosomal MLH1 foci, the frequency of XY separation (PAR untouching) in the mid-pachytene stage, and the frequency of MLH1 focus at the PAR are all comparable between *S100pbp*^{+/-} and wild-type mice (Appendix Fig. S2 and Fig A below).

In addition, we have specified the genotypes of the controls in the Material and Methods section (lines 621-623).

Fig A. The analysis of MLH1 focus in *S100pbp*^{+/-} mice. (i) Representative spread spermatocytes stained for MLH1 (green), SYCP3 (red), and H1t (grey). Miniaturized H1t staining (grey), shown in the lower-left corner of the overlay images, was used to identify the mid-pachytene spermatocytes (H1t-moderate). Scale bars, 10 μ m. (ii) A scatter plot showing the number of MLH1 foci on autosomes per nucleus. (iii) The frequency of nuclei with untouching XY PARs at the mid-pachytene stage. (iv) The frequency of nuclei with an MLH1 focus at the PAR. For (ii-iv), n indicates the number of nuclei scored from two mice per genotype.

5. 154: as bivalents (plural).

Response: Thank you for your careful reading. Corrected.

6. 165: fig 1H, n= ?

Response: The information is indicated in the figure legend for Fig. 1I, which provides the quantification of the results shown in Fig. 1H. We have revised the sentence to improve clarity.

7. Lines 175-195: The PAR with MLH1/3 foci counted for XY bivalents, or for all XY including separated ones? Because the proportions of bivalent XY (Fig. 1I) and of XY with a MLH1/3 focus (Fig. 2B,F) are similar, I wonder whether virtually every bivalent XY has a MLH1/3 focus in *S100pbp*^{-/-} spermatocytes. Please clarify and indicate whether there are MLH1/3 foci on PAR of separated XY.

Response: Thank you for the question. Taking into account the suggestions of Reviewer #1 (comment 2) and the Reviewer #2 (comment 17), we re-quantified the number of MLH1/3 foci focusing at the mid-pachytene stage, counting all XY regardless of whether PARs are touched or not. There are 67% and 74% of *S100pbp*^{+/-} mid-pachytene cells

lacking a MLH1 and MLH3 focus respectively (Fig. 2B and F), which are slightly higher than the frequency of XY PARs untouched (65%, Fig. EV1E).

As you suggested, we checked the presence of MLH1/3 on the bivalent XY. In *S100pbp*^{-/-} mice at the mid-pachytene stage, not virtually every bivalent XY has a MLH1/3 focus: 25 of the 31 bivalent XY harbor a MLH1 focus and 17 of the 21 bivalent harbor a MLH3 focus. No MLH1/3 foci were seen on XY chromosomes with the PARs untouched (49 and 34 nuclei analyzed for MLH1 and MLH3 respectively).

8. 183: Please indicate the age of the mice examined for MLH1 (and other) counts. Spermatocytes from adult and juvenile mice have been reported to have different MLH1-MLH3 (and RPA2) counts (Zelazowski et al., Cell, 2017), therefore age differences might alter the results.

Response: Thank you for the question. We used adult mice aged between 8 weeks and 12 weeks, except for the fertility test (we used mice of 8-10 weeks old). This was indicated in the Materials and Methods section (lines 515-516 and Appendix Table S1).

9. Lines 220-223: The counts of nuclei with RPA2 and RAD51 are shown only at the PAR of spermatocytes "with XY touching" on Fig. EV3 E-H. However, it is written in the main text that these counts are "comparable" for spermatocytes "with XY separated or touching". The counts of PAR of separated XY with RPA2 and RAD51 foci must also be shown. The criteria used to determine whether foci overlap the PAR must be indicated (e.g., PAR FISH, or specific distance measured from X or Y axis end).

Response: Thank you for pointing this out. We apologize for the confusion caused. We compared the frequencies of nuclei with RPA2 foci and RAD51 foci detected at the PAR between those with XY PARs touching and untouched in *S100pbp*^{-/-} mice. The results indicate no significant difference, suggesting that the PAR DSB repair is completed in the mutant spermatocytes, despite XY PARs being untouched. These results are now presented in Fig. EV2 H and J and are detailed in the results section (lines 256-257).

In our previous study on RAD51AP2, a RAD51-interacting protein that specifically promotes XY crossing over (Ma et al., 2022, Sci Adv), we conducted PAR FISH analysis to measure the ratio of PAR axis length to X or Y axis length at the early, mid, and late pachytene stages (see Fig. B below). This measurement enabled us to define the range of the PAR. We applied this criterion to assess whether a focus lies within the PAR (particularly for those with PAR untouched) and have reworded the sentences to improve clarity in the revised manuscript (lines 251-253).

Fig B. Determination of the relative axial length of PARs in early, mid, and late pachynema (Fig S16 from Ma et al, Sci Adv, 2022).

Reference:

Ma H, Li T, Xie X, et al., RAD51AP2 is required for efficient meiotic recombination between X and Y chromosomes. *Sci Adv.* 2022 Jan 14;8(2):eabk1789. doi: 10.1126/sciadv.abk1789.

10. 286: To identify S100PBP interactors, testicular lysates should have been compared between age-matched juvenile animals with similar cell composition: 10-week-old S100pbp^{-/-} testes are devoid of post-MI cells whereas WT testes are filled with haploid cells. Please show the list of peptides found in control IPs (IgG and S100pbp^{-/-}) on Table S2.

Response: Thank you for the comment. The list of peptides is shown in **Appendix Table S1**.

11. 412: the data showing that there is no change in chromosome length (SC in pachytene stage?) should be shown.

Response: Thank you for the comment. The data of the chromosome length is now presented in Appendix Fig S3B.

Again, we are very grateful to all of you for taking the time to review our manuscript. We sincerely hope that this revised manuscript has now addressed most of your concerns and meet with approval.

Hui

Dear Dr. Ma,

Thank you for submitting your revised manuscript. I have now read your point-by-point response carefully and I appreciate your response to the referee comments. However, I need you to address the points below as well before I can accept the manuscript.

- Please address the remaining concerns of referees #1 and #3 and provide a point-by-point response.
- Please provide 3-5 keywords for your study. These will be visible in the html version of the paper and on PubMed and will help increase the discoverability of your work.
- Please remove the "All data needed to support the conclusions in this study are presented in the paper and/or the Supplementary information files." sentence from the Data Availability section.
- Please remove the 'Author's contribution' section from the manuscript text.
- Please add a table of contents with page numbers to the title page of the Appendix file.
- All research articles submitted as revised versions must include a structured methods section that includes a Reagents and Tools Table followed by a Methods and Protocols section. Please see <https://www.embopress.org/page/journal/14693178/authorguide#structuredmethods> for further information.
- I would like to make the following suggestions regarding the Appendix tables given their contents and sizes:
 - o Please include Appendix Table S1 in the Appendix file.
 - o Please convert the Appendix Table S1 into Dataset EV1 and please update the source file name and its title in the manuscript submission system, the file, and the manuscript callouts.
 - o I note that Appendix Table S3 is a list of primer sequences used in the study, which should be a part of the Reagents and Tools Table (please see my above point). Similarly, please remember to update the callouts in the manuscript text.
- Materials and Methods should be renamed as Methods.
- Our production/data editors have asked you to clarify several points in the figure legends:
 - o Please note that information related to n is missing in the legends of figures 1A.
 - o Please note that n=2 in figures 4C, D.
 - o Please note that the error bars are not defined in the legends of figures 1A, I; 2B, C, D, F, G, H; 3B, C, E, F, H, I; 4C, D, F; EV1 B, D, E; EV2 C, D, G, H, I, J; EV5B, D, E, F; supplementary figure(s) 2B, C; 3B.
 - o Please note that the legends for figure 2, 3, EV2 is not provided in the sequential manner. This needs to be rectified.
 - o Please note that the exact p values are not provided in the legends of figures 1E, F, I; 2B, C, D, F, G, H; 3B, C, E, F, H; 4C, D; EV1 B, D, E; EV4 D, E; EV5 B, D, E, F.
 - o Please indicate the statistical test used for data analysis in the legends of supplementary figure(s) 2B, C; 3B.
 - o Please note that the letters (L, Z, EP, M/LP, D, Z) are not defined in the legend of figure 5C. This needs to be rectified.
 - o Please note that the letters (L, P) are not defined in the legend of figure 5D. This needs to be rectified."
- Papers published in EMBO Reports include a 'synopsis' and 'bullet points' to further enhance discoverability. Both are displayed on the html version of the paper and are freely accessible to all readers. The synopsis includes a short standfirst summarizing the study in 1 or 2 sentences (max 35 words) that summarize the paper and are provided by the authors and streamlined by the handling editor. I would therefore ask you to include your synopsis blurb and 3-5 bullet points listing the key experimental findings.
- In addition, please provide an image for the synopsis. This image should provide a rapid overview of the question addressed in the study but still needs to be kept fairly modest since the image size cannot exceed 550 (width) x 300-600 (height) pixels.

Thank you again for giving us to consider your manuscript for EMBO Reports, I look forward to your minor revision.

Kind regards,

Deniz Senyilmaz Tiebe

--

Deniz Senyilmaz Tiebe, PhD
Senior Scientific Editor
EMBO Reports

Referee #1:

The authors have adequately addressed all of my concerns **except for the last**, which involves conclusions drawn from analysis of the S100pbp Δ 177bp/ S100pbp Δ 177bp mice. As authors acknowledge, they cannot currently show that this mutant produces any stable protein at all, and thus the null-like phenotype of this mutant cannot be definitively ascribed to a loss of interaction with TPR. However, they do not say this explicitly until the discussion, after they have made two potentially misleading statements, one in the results (lines 399-401) and at the beginning of the discussion (lines 410-412).

I have no problem with including these data in the paper, as long as the following things are done:

First, the phrase "and S100pbp mutant mice carrying a mutation in the TPR-interacting domain display similar impairments in crossover formation as in S100pbp $^{-/-}$ mice, suggesting that S100PBP likely functions in a TPR-dependent manner" must be

removed from the abstract.

Second, edit the results/discussion section either of 2 ways:

1. Remove "which mediates the interaction with TPR" from line 400; remove the last sentence of the first paragraph of the discussion.

2. Alternatively, move the caveat text in the discussion (lines 482-495) to the end of the results section, so that the reader is informed at the time the data are presented of the constraints on interpreting the current results.

Frankly, I fail to understand why the authors insist on including the S100pbbp Δ 177bp data in the manuscript. As stated in my previous review, the manuscript is strong enough without it, and the S100pbbp Δ 177bp data, if combined with a demonstration that the mutant protein is present at levels similar to wild-type (either with a different antibody, or by engineering in an endogenous tag), would be the basis for a subsequent high-impact paper.

Referee #2:

My comments have been properly addressed.

Referee #3:

The authors have answered adequately to most enquiries, but I still have a few concerns. This manuscript is now appropriate for publication in EMBO Reports, provided that these concerns are considered.

Lines 18-129: What is the exact meaning of "any premature truncated protein"? This frameshift allele might still produce a truncated protein missing the N-ter region.

Lines 88, 445: Please cite also the Alavattam et al reference about Atf7ap2 mutant.

Line 91: mechanisms

Line 181: The numbers given in the point per point response (comment 7) result in percentages of nuclei with no MLH1/3 focus on the PAR that are slightly different from the ones in the text (MLH1: $(6+49)/80=68.75\%$, MLH3: $(4+34)/55=69\%$). Please correct them.

Line 308: Since co-IP MS experiments were done in 10-week old mice rather than juveniles, the fact that WT and mutant mice have different cell composition must be mentioned explicitly, as it implies that the significance of differences between WT and mutant is deeply altered.

Line 319: In Fig EV3B, the control for GFP-TPR IP should be no GFP-TPR instead of with no Flag-S100PBP.

Here are our point-by-point responses to reviews:

We would like to thank you again for taking the time and effort necessary to review our manuscript. We sincerely appreciate all your valuable comments and suggestions, which have greatly helped us to improve the quality of the manuscript.

Referee #1:

The authors have adequately addressed all of my concerns except for the last, which involves conclusions drawn from analysis of the S100pbp Δ 177bp/ S100pbp Δ 177bp mice. As authors acknowledge, they cannot currently show that this mutant produces any stable protein at all, and thus the null-like phenotype of this mutant cannot be definitively ascribed to a loss of interaction with TPR. However, they do not say this explicitly until the discussion, after they have made two potentially misleading statements, one in the results (lines 399-401) and at the beginning of the discussion (lines 410-412).

I have no problem with including these data in the paper, as long as the following things are done:

First, the phrase "and S100pbp mutant mice carrying a mutation in the TPR-interacting domain display similar impairments in crossover formation as in S100pbp $^{-/-}$ mice, suggesting that S100PBP likely functions in a TPR-dependent manner" must be removed from the abstract.

Second, edit the results/discussion section either of 2 ways:

1. Remove "which mediates the interaction with TPR" from line 400; remove the last sentence of the first paragraph of the discussion.
2. Alternatively, move the caveat text in the discussion (lines 482-495) to the end of the results section, so that the reader is informed at the time the data are presented of the constraints on interpreting the current results.

Frankly, I fail to understand why the authors insist on including the S100pbp Δ 177bp data in the manuscript. As stated in my previous review, the manuscript is strong enough without it, and the S100pbp Δ 177bp data, if combined with a demonstration that the mutant protein is present at levels similar to wild-type (either with a different antibody, or by engineering in an endogenous tag), would be the basis for a subsequent high-impact paper.

Response: We sincerely appreciate your insightful suggestion and constructive feedback. We have removed S100pbp Δ 177bp data and the related discussion from the manuscript.

Referee #2:

My comments have been properly addressed.

Referee #3:

The authors have answered adequately to most enquiries, but I still have a few concerns. This manuscript is now appropriate for publication in EMBO Reports, provided that these concerns are considered.

Lines 18-129: What is the exact meaning of "any premature truncated protein"? This frameshift allele might still produce a truncated protein missing the N-ter region.

Response: Thank you for your careful reading. We did not mention any truncated protein missing the N-ter region (initiating from an internal M at amino acid 60) in this context, because even if such a truncated form were present, it would not be detected by our anti-S100PBP antibody (the epitope: amino acids 19-33 of the mouse S100PBP protein). Upon reviewing your comment, we realized that we also lack sufficient evidence that the premature truncated protein lacking the C-terminus (p.G27Ffs*4) could be recognized by the anti-S100PBP antibody. Therefore, we have revised the text to: "Western blotting detected the presence of full-length S100PBP in the wild-type testes, but not in *S100pbp*^{-/-} testes. It should be noted that the frameshift allele may produce a truncated protein (p.G27Ffs*4) lacking the C-terminal region or a truncated protein missing the N-terminal region due to translation reinitiation, which, if present, may not be recognized by the anti-S100PBP antibody" (lines 124-129).

Lines 88, 445: Please cite also the Alavattam et al reference about Atf7ap2 mutant.

Response: Thank you for your careful reading. We have cited the reference at these two places.

Line 91: mechanisms

Response: Thank you for your careful reading. Corrected.

Line 181: The numbers given in the point per point response (comment 7) result in percentages of nuclei with no MLH1/3 focus on the PAR that are slightly different from the ones in the text (MLH1: (6+49)/80=68.75%, MLH3: (4+34)/55=69%). Please correct them.

Response: Thank you for your careful observation. The percentage of nuclei lacking an MLH1 or MLH3 focus on the PAR (67% and 74%, respectively) were calculated as averages from three independent experiments. In contrast, the number of nuclei lacking an MLH1 or MLH3 focus on the PAR (6+49 and 4+34, respectively) represent the raw counts from those three independent experiments. This explains the slight discrepancy between the percentages reported in the text and the calculated values based on the provided numbers.

The raw data from the three independent experiments are shown here for reference.

	MLH1		MLH3	
	Control	S100pbp-/-	Control	S100pbp-/-
Experiment 1 (%)	92.86	28.57	85.71	32.35
Experiment 2 (%)	90.48	34.38	83.333	20
Experiment 3 (%)	92.31	35	93.75	25
Average (%)	91.88	32.65	87.60	25.78
Total number of nuclei scored from the three experiments	48	80	43	55

*These raw data have been uploaded as the Source Data.

Line 308: Since co-IP MS experiments were done in 10-week old mice rather than juveniles, the fact that WT and mutant mice have different cell composition must be mentioned explicitly, as it implies that the significance of differences between WT and mutant is deeply altered.

Response: Thank you for your valuable and constructive feedback. We have mentioned this in the revised manuscript (lines 305-308).

Line 319: In Fig EV3B, the control for GFP-TPR IP should be no GFP-TPR instead of with no Flag-S100PBP.

Response: Thank you for your valuable and constructive suggestion. We have re-performed the experiment as you suggested and updated the Fig EV3B (also shown below) in the current version of manuscript.

Fig EV3B. Co-IP was performed using an anti-GFP antibody in HEK293T cells that were exogenously expressing S100PBP (with an N-terminal Flag tag) and TPR (with an N-terminal GFP tag), followed by western blotting with the anti-Flag and anti-GFP antibodies. The results show that Flag-S100PBP was detected in the lysate immunoprecipitated by the GFP beads when co-transfected with GFP-TPR, but not in the absence of GFP-TPR.

Again, we are sincerely grateful to all of you for taking the time to review our manuscript. We sincerely hope that this revised manuscript has now addressed most of your concerns and meet with approval.

Hui

Hui Ma
University of Science and Technology of China
The First Affiliated Hospital of USTC
Huangshan Road 443
Hefei, Not Applicable 230027
China

Dear Dr. Ma,

Thank you for submitting your revised manuscript. I have now looked at everything and all is fine. Therefore, I am very pleased to accept your manuscript for publication in EMBO Reports.

Congratulations on a nice work!

Kind regards,

Deniz Senyilmaz Tiebe

--

Deniz Senyilmaz Tiebe, PhD
Senior Scientific Editor
EMBO Reports
